# Improving word mover's distance by leveraging self-attention matrix

**Hiroaki Yamagiwa**[1]    **Sho Yokoi** [2,3]    **Hidetoshi Shimodaira** [1,3]
[1]Kyoto University    [2]Tohoku University    [3]RIKEN AIP
hiroaki.yamagiwa@sys.i.kyoto-u.ac.jp,
yokoi@tohoku.ac.jp, shimo@i.kyoto-u.ac.jp

## Abstract

Measuring the semantic similarity between two sentences is still an important task. The word mover's distance (WMD) computes the similarity via the optimal alignment between the sets of word embeddings. However, WMD does not utilize word order, making it challenging to distinguish sentences with significant overlaps of similar words, even if they are semantically very different. Here, we attempt to improve WMD by incorporating the sentence structure represented by BERT's self-attention matrix (SAM). The proposed method is based on the Fused Gromov-Wasserstein distance, which simultaneously considers the similarity of the word embedding and the SAM for calculating the optimal transport between two sentences. Experiments demonstrate the proposed method enhances WMD and its variants in paraphrase identification with near-equivalent performance in semantic textual similarity. Our code is available at https://github.com/ymgw55/WSMD.

## 1  Introduction

The task of measuring the semantic textual similarity (STS) of two sentences is essential for natural language processing with various applications (Cer et al., 2017). There are several methods for measuring STS, among which many methods using Optimal Transport (OT) distance have been proposed and have shown good performance (Kusner et al., 2015; Huang et al., 2016; Chen et al., 2019; Yokoi et al., 2020).

OT theory gives a method to measure the difference between two distributions by setting the transport cost of a unit mass and considering the allocation of the transported mass to minimize the total cost. This allocation is called optimal transport, and the total cost is called the OT distance.

A basic form of OT distance is the Wasserstein distance, which measures the similarity between sets using the distance between the elements in

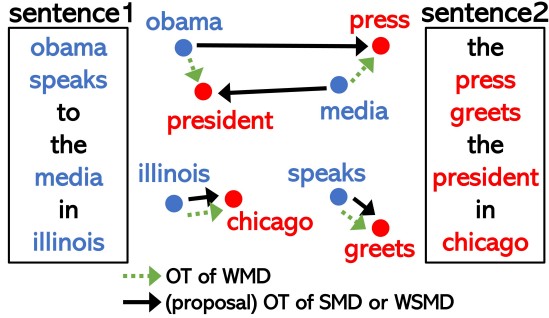

Figure 1: An illustration of OT for word embeddings from sentence 1 to sentence 2. Words are aligned by word similarity in WMD; e.g., *obama* matches *president*. Words are aligned by sentence structure in SMD or by word similarity and sentence structure simultaneously in WSMD; e.g., *obama* matches *press*. See Section 2.

the sets (Kantorovich, 1960). Word Mover's Distance (WMD) computes Wasserstein distance by considering a sentence as a set of word embeddings (Kusner et al., 2015).

However, WMD does not consider the word order of sentences, making it challenging to identify paraphrases. Let us see the following illustrative example of the paraphrases adapted from (Kusner et al., 2015) and (Zhang et al., 2019):

(a) Obama speaks to the media in Illinois.

(b) The President greets the press in Chicago.

(c) The press greets the President in Chicago.

Here (b) is a paraphrase of (a), while (c) has a very different meaning from (a). However, all these sentences have a high overlap of words. Thus, WMD cannot distinguish these pairs.

To account for sentence structure, we focused on BERT (Devlin et al., 2019), an attention-based model that has recently achieved remarkable performance in natural language processing tasks. BERT is a masked language model that performs task-specific fine-tuning after pre-training on a large

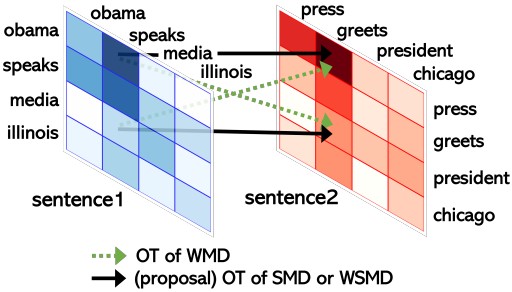

Figure 2: An illustration of OT for SAMs from sentence 1 to sentence 2. To avoid crowding the diagram, arrows are shown in only two SAM elements. The darker the color, the higher the value of the SAM element (reflecting a real SAM used in Table 1). The difference of SAM values is small for the word alignment of SMD or WSMD; e.g., (*obama*, *speaks*) matches (*press*, *greets*). The difference is large for that of WMD; e.g., (*obama*, *speaks*) matches (*president*, *greets*). See Section 2.

dataset. By inputting a sentence into the pretrained BERT, we can extract the Self-Attention Matrix (SAM), which represents the dependencies between words in the sentence. SAM encodes sentence structure information, such as syntactic information (Clark et al., 2019; Htut et al., 2019; Luo, 2021). Therefore, sentence structure can be incorporated into STS by measuring the distance between SAMs.

We propose a novel method called sentence Structure Mover's Distance (SMD) in Section 5 that measures the optimal transport distance between sentence structures represented in SAMs. SMD aims to find the optimal word alignment by considering the transport cost between elements of SAMs, while WMD considers the transport cost between word embeddings. To compute SMD, we employ the Gromov-Wasserstein (GW) distance (Mémoli, 2011; Peyré and Cuturi, 2020), the optimal transport distance for measuring structural similarity between sets.

SMD is ineffective by itself because it measures only the difference between sentence structures. Thus SMD is used in combination with WMD. We improve WMD-like methods (i.e., WMD and its variants based on Wasserstein distance between sets of word embeddings) by combining SMD with them in Section 6. This proposed method is called Word and sentence Structure Mover's Distance (WSMD). WSMD simultaneously measures the difference between word embeddings and the difference between sentence structures. WSMD

employs the Fused Gromov-Wasserstein (FGW) distance (Vayer et al., 2019, 2020), known as the optimal transport distance that combines both Wasserstein and GW distances.

## 2 Illustrative example

Here we explain how our proposed method works for the example of the paraphrases in Section 1. The original WMD and its improvement with our proposal are computed for this example and shown in Table 1. We computed these values using the word embeddings taken from the input layer of BERT and the SAMs taken from the second head of the eighth layer of BERT. The details of the setting will be described in Section 7.2. The similarity should be high for the sentence pair (a) vs. (b), i.e., case (1), while the similarity should be low for the sentence pair (a) vs. (c), i.e., case (2).

Looking at the values of the original WMD for the two cases, we find that they are about the same. Thus, WMD failed to give a reasonable sentence similarity. This failure is explained in Fig. 1 by illustrating word embeddings for case (2). WMD tries to find the closest word for each word, matching *obama* to *president*, *media* to *press*, and so on; the word alignment is indicated as the short dotted arrows. Then, WMD is computed by averaging the length of these short arrows, indicating a high sentence similarity contrary to the fact that the actual similarity is low for case (2).

However, the word alignment for WMD is inappropriate, given the word order and sentence structure. Our proposed method correctly matches *obama* to *press* and *media* to *president*. This word alignment is shown as the long arrows with solid lines in Fig. 1. $WMD_\lambda$ in Table 1 is computed by averaging the length of these arrows, i.e., the same formula as WMD but using the OT taking account of the sentence structure. We find that $WMD_\lambda$ increases for case (2), correctly indicating a low sentence similarity.

Fig. 2 explains how our proposed method takes account of the sentence structure. Given a word alignment between two sentences, we can think of a matching of elements between the two SAMs by applying the word alignment to both the rows and the columns of the matrices. For example, the word alignment of WMD in Fig. 1 induces the matching from (*obama*, *speaks*) to (*president*, *greets*), (*media*, *speaks*) to (*press*, *greets*), and so on. SMD minimizes the average difference of SAM values

| case | sentences | WMD | WMD$_\lambda$ | WSMD | similarity |
|------|-----------|-----|--------------|------|-----------|
| (1) | (a) obama speaks to the media in illinois. (b) the president greets the press in chicago. | 12.54 | 12.54 | 7.26 | high |
| (2) | (a) obama speaks to the media in illinois. (c) the press greets the president in chicago. | 12.55 | 13.30 | 8.03 | low |

Table 1: Similarity measures for sentence pairs (a) vs. (b) and (a) vs. (c) of Section 1.

between the matched elements. This recovers the correct word alignment in this example. WSMD minimizes the weighted sum of the objectives for WMD and SMD; we used the mixing ratio $\lambda = 0.5$ here. As shown in Table 1, WSMD correctly indicates that the sentences in case (2) are less similar than those in case (1).

## 3 Related Work

This paper focuses on sentence similarity measures using the OT of word embeddings. For example, WMD (Kusner et al., 2015) uses uniform word weights and the $L_2$ distance of word embeddings as the transport cost. By modifying the word weights and the transport cost, various WMD-like methods have been developed to compute sentence similarity based on the Wasserstein distance between sets of word embeddings. For example, Word Rotator's Distance (WRD) (Yokoi et al., 2020) uses the vector norm of the word embedding for the weight and the cosine similarity for the cost. There are many attempts to improve WMD by incorporating word order and sentence structure information into the weight, cost, and penalty terms, as explained below. However, none of them consider the optimal transport of sentence structures nor utilize SAMs from BERT.

In OPWD (Su and Hua, 2019), the temporal difference between word embeddings $\mathbf{w}_i$ and $\mathbf{w}'_j$ is measured by the distance between their relative temporal positions $(i/n - j/m)^2$, where $n$ and $m$ represent the lengths of the corresponding sentences. This difference is incorporated into the transport cost and an additional regularization term. WMDo (Chow et al., 2019) identifies consecutive words common to two sentences as a chunk and introduces a penalty term according to the number of chunks. In Syntax-aware WMD (SynWMD) (Wei et al., 2022), the word weight is computed from the word co-occurrence extracted from the syntactic parse tree of sentences, and a word embedding incorporates those from its subtree of the parse tree.

The weight and the cost in SynWMD are called Syntax-aware Word Flow (SWF) and Syntax-aware Word Distance (SWD), respectively. In Recursive Optimal Transport Similarity (ROTS) (Wang et al., 2020, 2022), a recursive optimal transport method is introduced to capture word dependencies in sentences. To measure sentence similarity, the pairwise cosine similarity of words is weighted using this optimal transport and aggregated. In Mover-Score (Zhao et al., 2019), the inverse document frequency (IDF) is used for the word weight, and the word embeddings from BERT are used for computing the cost. BERTscore (Zhang et al., 2020a) also uses IDF and BERT embedding, but it employs greedy matching instead of OT for word alignment.

## 4 Optimal transport of words

We review the computation of WMD. In the following, we denote two sentences as

$$s = (\mathbf{w}_i)_{i=1}^n, \ s' = (\mathbf{w}'_j)_{j=1}^m \subset \mathbb{R}^d,$$

where $\mathbf{w}_i, \mathbf{w}'_j \in \mathbb{R}^d$ are word embeddings, and $n, m$ are sentence lengths. SAM is denoted $\mathbf{A} = (A_{ii'}) \in \mathbb{R}^{n \times n}$ and $\mathbf{A}' = (A'_{jj'}) \in \mathbb{R}^{m \times m}$, respectively, for $s$ and $s'$. The element $A_{ii'}$ is the attention weight from $\mathbf{w}_i$ to $\mathbf{w}_{i'}$, and the element $A'_{jj'}$ is the attention weight from $\mathbf{w}'_j$ to $\mathbf{w}'_{j'}$. Each row of SAM is normalized as $\sum_{i'=1}^n A_{ii'} = \sum_{j'=1}^m A'_{jj'} = 1$.

The weights on $\mathbf{w}_i$ and $\mathbf{w}'_j$ in the probability distributions for $s$ and $s'$ are denoted by $\mathbf{u} = (u_i)_{i=1}^n \in \mathbb{R}_{\geq 0}^n$, with $\sum_{i=1}^n u_i = 1$, and $\mathbf{u}' = (u'_j)_{j=1}^m \in \mathbb{R}_{\geq 0}^m$, with $\sum_{j=1}^m u'_j = 1$, respectively. In this paper, unless otherwise stated, the weights on words in a sentence are equal, i.e., the uniform distribution specified as

$$\mathbf{u} = (1/n)_{i=1}^n, \ \mathbf{u}' = (1/m)_{j=1}^m. \tag{1}$$

### 4.1 Wasserstein distance

Sentences $s$ and $s'$ cannot be naively compared because they are generally different in length, and the corresponding words are unknown. We then consider the word transport from $s$ to $s'$ to obtain

the word alignment. Interpret the sentence $s$ as the amount of mass $u_i$ at position $\mathbf{w}_i$. First, we denote $P_{ij} \in [0, 1]$, the amount of mass transported from position $\mathbf{w}_i$ to position $\mathbf{w}'_j$, and consider the transport matrix $\mathbf{P} = (P_{ij}) \in \mathbb{R}^{n \times m}_{\geq 0}$ with nonnegative elements. Next, we define the distance function $c(\mathbf{w}_i, \mathbf{w}'_j) \in \mathbb{R}_{\geq 0}$ as the cost of transporting a unit mass from $\mathbf{w}_i$ to $\mathbf{w}'_j$ and specify the distance matrix $\mathbf{C} = (C_{ij}) = (c(\mathbf{w}_i, \mathbf{w}'_j)) \in \mathbb{R}^{n \times m}_{\geq 0}$. Given $\mathbf{P}$ and $\mathbf{C}$, the transport from $\mathbf{w}_i$ to $\mathbf{w}'_j$ costs $C_{ij}P_{ij}$, and the total cost of transport is

$$\sum_{i=1}^{n} \sum_{j=1}^{m} C_{ij} P_{ij}. \quad (2)$$

Finding the optimal transport matrix $\hat{\mathbf{P}} = (\hat{P}_{ij})$ that minimizes the total cost, we compute the Wasserstein distance between $s$ and $s'$ as the minimum value of the total cost $\sum_{i=1}^{n} \sum_{j=1}^{m} C_{ij} \hat{P}_{ij}$.

## 4.2 Word mover's distance

Word Mover's Distance (WMD) (Kusner et al., 2015) is Wasserstein distance applied to word embeddings. In the original WMD, the weight on words is the uniform distribution (1), and the distance function is Euclid distance between word embeddings $\mathbf{w}_i$ and $\mathbf{w}'_j$. The distance matrix is defined as

$$C_{ij} = c(\mathbf{w}_i, \mathbf{w}'_j) = \|\mathbf{w}_i - \mathbf{w}'_j\|_2. \quad (3)$$

Therefore, WMD between $s$ and $s'$ is

$$\text{WMD}(s, s') = \min_{\mathbf{P} \in \mathbf{\Pi}(\mathbf{u}, \mathbf{u}')} \sum_{i=1}^{n} \sum_{j=1}^{m} C_{ij} P_{ij}. \quad (4)$$

Here, $\mathbf{\Pi}(\mathbf{u}, \mathbf{u}')$ is the set of all possible values of the transport matrix $\mathbf{P}$ defined as

$$\left\{ \mathbf{P} \in \mathbb{R}^{n \times m}_{\geq 0} \mid \sum_{i=1}^{n} P_{ij} = u'_j, \sum_{j=1}^{m} P_{ij} = u_i \right\},$$

and $\mathbf{u}, \mathbf{u}'$ are omitted on the left side of (4).

## 4.3 Limitations of WMD

WMD cannot account for the word order of the sentences because it treats a sentence as a set of word embeddings to find the optimal transport distance. Therefore, it is hard to distinguish sentence pairs with significant word overlaps in the case of static word embeddings. For some models, such as BERT and ELMo, the problem remains even for dynamic word embeddings that depend on the context because the similarity is still high between word embeddings of the same word (Ethayarajh,

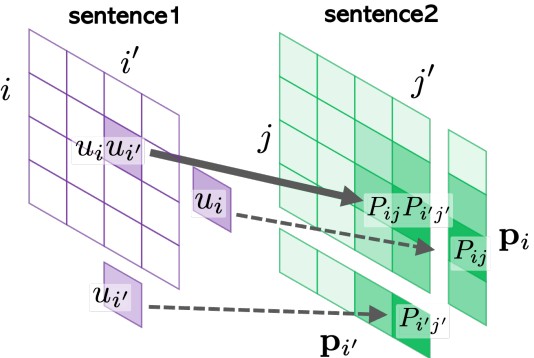

Figure 3: Transport of sentence structure induced by word transport. The array of transport from the $i$-th word is $\mathbf{p}_i$ and that from the $i'$-th word is $\mathbf{p}_{i'}$. The amount of transport from the $(i, i')$-th element of SAM is defined by the outer product of $\mathbf{p}_i$ and $\mathbf{p}_{i'}$.

2019). WMD does not distinguish the case (1) and the case (2) of Table 1 in Section 2 using the 0th layer of BERT as a statistic word embedding. Moreover, the difference is slight even if the 12th layer of BERT is used as a dynamic word embedding: WMD = 9.17 for case (1) and WMD = 9.26 for case (2).

## 5 Optimal transport of sentence structures

In this section, we propose to apply the GW distance to the SAM of BERT to measure the optimal transport distance between sentence structures. Later in Section 6, we will attempt to improve WMD using this method.

## 5.1 Gromov-Wasserstein distance

As seen in Section 2, the transport of sentence structure from $s$ to $s'$, i.e., transport from $\mathbf{A}$ to $\mathbf{A}'$, is induced by the transport of words. How can we define the structure transport consistent with a given word transport?

Let us consider $\mathbf{p}_i = (P_{i1}, \ldots, P_{im})$, the array of transport amount $P_{ij}$ from the $i$-th word of $s$ to the $j$-th word of $s'$. As illustrated in Fig. 3, we apply $\mathbf{p}_i$ to the $i$-th row of $\mathbf{A}$ and $\mathbf{p}_{i'}$ to the $i'$-th column of $\mathbf{A}$. This defines the transport from $A_{ii'}$ of $s$ to $A'_{jj'}$ of $s'$ as the outer product of $\mathbf{p}_i$ and $\mathbf{p}_{i'}$; the transport amount from position $A_{ii'}$ to position $A'_{jj'}$ is defined as $P_{ij}P_{i'j'}$.

The above definition of transport between SAMs is consistent with the transport of words. From the definition of transport, the mass at $A_{ii'}$ is $\sum_{j,j'=1}^{m} P_{ij}P_{i'j'} = u_i u_{i'}$, and the mass at $A_{jj'}$ is

$\sum_{i,i'=1}^{n} P_{ij}P_{i'j'} = u'_j u'_{j'}$. This implies that the marginal mass for rows and columns coincides with the mass of words, and thus the transport preserves the mass.

In this paper, we specify the cost of transporting a unit mass as $|A_{ii'} - A'_{jj'}|^2$. Then, the cost of transport is $|A_{ii'} - A'_{jj'}|^2 P_{ij}P_{i'j'}$, and the total cost of transport from $\mathbf{A}$ to $\mathbf{A}'$ is

$$\mathcal{E}_{\mathbf{A},\mathbf{A}'}(\mathbf{P}) := \sum_{i,i'=1}^{n} \sum_{j,j'=1}^{m} |A_{ii'} - A'_{jj'}|^2 P_{ij}\, P_{i'j'} \tag{5}$$

Finding the optimal transport matrix $\hat{\mathbf{P}}$ that minimizes the total cost $\mathcal{E}_{\mathbf{A},\mathbf{A}'}(\mathbf{P})$, we compute the Gromov-Wasserstein (GW) distance (Mémoli, 2011; Peyré and Cuturi, 2020) between $\mathbf{A}$ and $\mathbf{A}'$ as the minimum value of (5).

## 5.2 Sentence structure mover's distance

Applying the GW distance to SAMs, we propose the sentence Structure Mover's Distance (SMD), which is the optimal transport distance considering the dependency of words in a sentence.

$$\text{SMD}(\mathbf{A}, \mathbf{A}') = \min_{\mathbf{P} \in \mathbf{\Pi}(\mathbf{u},\mathbf{u}')} \mathcal{E}_{\mathbf{A},\mathbf{A}'}(\mathbf{P}) \tag{6}$$

Note that SMD is not a metric of a metric space, while it can still measure the structural difference. The general definition of GW distance considers the $p$-th root of the total cost for the form $|A_{ii'} - A'_{jj'}|^p$ in (5), and it is a metric if $A_{ii'}$ is a symmetric distance matrix. Thus using an asymmetric SAM is not GW distance in a strict sense. We consider the case $p = 2$ and the square root is omitted because it does not affect the magnitude order.

## 6 Optimal transport of words and sentence structures

Here, we propose an optimal transport distance that simultaneously considers the word embeddings and sentence structure.

### 6.1 Word and sentence structure mover's distance

As seen in Section 4.3, WMD computes Wasserstein distance using Euclid distance of word embeddings but cannot handle sentences with different meanings depending on word order. On the other hand, as seen in Section 5.2, SMD computes the GW distance using the SAM of BERT, which encodes the sentence structure, but it cannot handle individual word information like word embeddings.

Therefore, by combining WMD-like methods (i.e., WMD and its variants obtained by modifying $\mathbf{C}$, $\mathbf{u}$ and $\mathbf{u}'$) with SMD, we propose an optimal transport distance, Word and sentence Structure Mover's Distance (WSMD), that utilizes word features and considers word dependency within a sentence. By specifying the mixing ratio parameter $\lambda \in [0, 1]$, we obtain

$$\text{WSMD}((s, \mathbf{A}), (s', \mathbf{A}')) =$$
$$\min_{\mathbf{P} \in \mathbf{\Pi}(\mathbf{u},\mathbf{u}')} \sum_{i,i'=1}^{n} \sum_{j,j'=1}^{m} \Big\{ (1-\lambda)C_{ij}$$
$$+ \lambda k \left| A_{ii'} - A'_{jj'} \right|^2 \Big\} P_{ij}P_{i'j'}, \tag{7}$$

where $k$ is a normalization parameter. For further details about $k$, refer to Appendix A. By noting $\sum_{i'=1}^{n} \sum_{j'=1}^{m} P_{i'j'} = 1$, WSMD = WMD for $\lambda = 0$. For $\lambda = 1$, WSMD = $k$SMD.

For an intermediate value $\lambda \in (0, 1)$, WSMD considers both WMD and SMD. Let $\hat{\mathbf{P}}$ be the optimal transport matrix, i.e., $\mathbf{P}$ that attains the minimum of (7). Substitute this $\hat{\mathbf{P}}$ into (2) and (5), denote them as $\text{WMD}_\lambda$ and $\text{SMD}_\lambda$, respectively. Then we can write

$$\text{WSMD} = (1-\lambda)\text{WMD}_\lambda + \lambda k \text{SMD}_\lambda. \tag{8}$$

This decomposition is explained in Appendix B.

The optimal transport distance that simultaneously considers the Wasserstein and GW distances, as in WSMD, is known as the Fused Gromov-Wasserstein (FGW) distance (Vayer et al., 2019, 2020). However, like SMD, WSMD uses an asymmetric SAM, so WSMD is not a metric in general.

## 7 Experiments

We compare the performance of our proposed method and existing baseline methods on the task of measuring semantic textual similarity (STS) between two sentences.

### 7.1 Datasets

#### 7.1.1 PAWS dataset

Paraphrase Adversaries from Word Scrambling (PAWS) (Zhang et al., 2019) has a binary label for an English sentence pair indicating paraphrase (i.e., the sentence pair has the same meaning) or non-paraphrase. The binary classification of these labels can be considered an STS task. If the transport distance for a pair is small, we classify the pair as a paraphrase; otherwise, we classify it as a non-paraphrase. The effectiveness of this classification

| | Dev | Test |
|---|---|---|
| PAWS$_{QQP}$ | 1500 (24.7%) | 677 (28.2%) |
| PAWS$_{Wiki}$ | 1500 (42.3%) | 8000 (44.2%) |
| STSB | 1500 | 1379 |

Table 2: The number of sentence pairs used in our experiments. Train sets are not used. The percentage of paraphrases for PAWS is shown in parentheses.

is then evaluated using the AUC as the metric.

There are two types of PAWS: PAWS$_{QQP}$ and PAWS$_{Wiki}$, constructed from sentences in Quora and Wikipedia, respectively. Table 2 shows the number of sentence pairs and the percentage of paraphrases used in our experiments. In PAWS$_{QQP}$, since the test set is not provided, the first 1500 pairs were selected from the 11988 pairs in the train set to make a new dev set, and the original dev set was considered as a test set. In PAWS$_{Wiki}$, the first 1500 pairs were selected from the dev set.

### 7.1.2 STS Benchmark dataset

The STSB dataset (Cer et al., 2017) contains human-annotated scores for English sentence pairs, reflecting the average similarity on a six-point scale. Table 2 shows the number of sentence pairs. Spearman's rank correlation (Spearman's $\rho$) is used to compare these scores with optimal transport distances.

### 7.2 Models

We used BERT, SimCSE (Gao et al., 2021) and RoBERTa (Liu et al., 2020) from the Hugging Face transformers library (Wolf et al., 2020) and see Table 8 in Appendix C for details.

SimCSE is a sentence embedding model based on contrastive learning with BERT. It has two types: unsup-SimCSE and sup-SimCSE, depending on how positive samples are defined. In this study, unsup-SimCSE is used.

RoBERTa is a model that improves the performance of BERT by changing the pre-training settings of BERT, and by tuning the hyperparameters.

The BERT, SimCSE, and RoBERTa models we used have 12 layers and 12 heads, resulting in a total of 144 extractable SAMs. We input the sentences into BERT and removed the stopwords and special tokens such as [CLS] and [SEP] from the output tokens[1].

The word embeddings used in the experiments are the static embeddings[2] taken from the input layer, i.e., the 0th layer, and the dynamic embeddings taken from the final layer, i.e., the 12th layer. We call such word embeddings BERT0, BERT12, and so on. When computing embeddings for STS methods such as WMD, recomputing the embeddings for each method may result in different embeddings due to the randomness of dropout. Therefore, to ensure an accurate comparison of STS methods, we had computed the embeddings once and used them for all the STS methods. We performed whitening for all the word embeddings because the representation of BERT is known to be anisotropic (Ethayarajh, 2019).

### 7.3 Baseline methods

The following simple baseline methods for representing a sentence were selected. The similarity for a sentence pair is computed by the cosine similarity of the two vectors.

**Bag-of-Words (BoW)** is a high-dimensional vector whose elements are the frequency of occurrence in a sentence for all words.

**Average Pooling (Avg. Pool.)** is simply the average vector of word embeddings in a sentence.

**[CLS] token**[3] is the first token in the input sequence, and in SimCSE it is used specifically for sentence embedding.

**SIF** (Arora et al., 2017) is an unsupervised method that sums word embeddings with frequency-based weights and subtracts the first singular vector from the SVD.

**uSIF** (Ethayarajh, 2018) is an unsupervised method that replaces the dot product with arccosine in the probability model of SIF and subtracts multiple singular vectors instead of just the first.

**Conceptor Negation (Con. Neg.)** (Liu et al., 2019) is an unsupervised method that optimizes the performance of sentence embeddings for a new corpus while preserving the performance for older corpora when dealing with multiple datasets.

**BERTScore** (Zhang et al., 2020b) is an automatic evaluation metric for text generation and computes a similarity score by comparing each token in a sentence with each token in another sen-

---

[1] For one of the benchmarking OT methods (SynWMD), we did not remove the stopwords since removing them would have deteriorated the performance.

[2] Note that the embeddings taken from the 0th layer are only approximate statistic embeddings, because the segmentation embeddings and the position embeddings are added.

[3] Instead of [CLS] token, we used  token for RoBERTa.

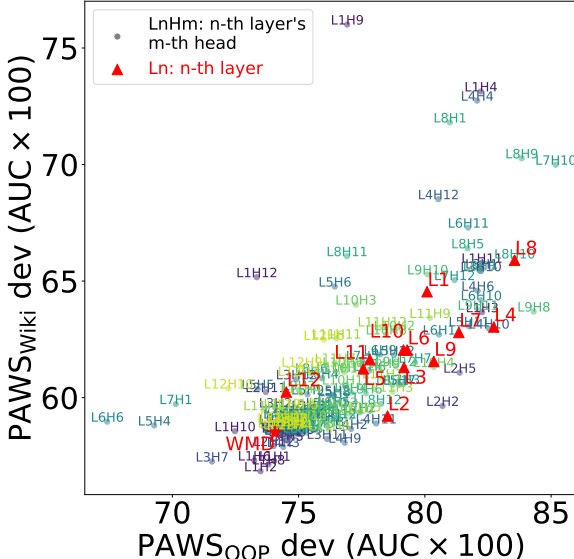

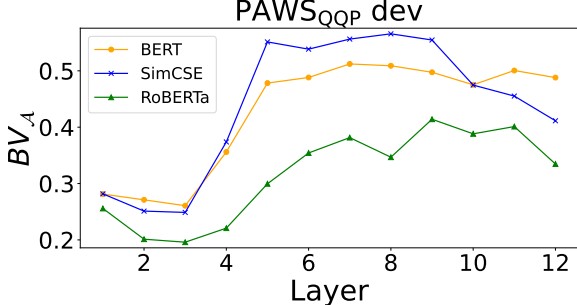

Figure 5: The layer-wise $\text{BV}_{\mathcal{A}}$ for BERT, SimCSE, and RoBERTa on the $\text{PAWS}_{\text{QQP}}$ dev set. $\text{BV}_{\mathcal{A}}$ is high from the fifth layer in each model.

| | BERT | SimCSE | RoBERTa |
|---|---|---|---|
| 0th Embs. | 8 | 6 | 9 |
| 12th Embs. | 8 | 6 | 9 |

Table 4: Top1 layer for BERT, SimCSE, and RoBERTa. The layer was selected from the fifth onward based on the AUC of the average WSMD for a single layer using the $\text{PAWS}_{\text{QQP}}$ dev set.

Figure 4: The AUC values of WMD ($\star$) and WSMD ($\bullet$, $\blacktriangle$) on the $\text{PAWS}_{\text{QQP}}$ (x-axis) and $\text{PAWS}_{\text{Wiki}}$ (y-axis) dev sets for BERT0. The $\bullet$ symbol indicates the AUC value of the WSMD for a single head. The $\blacktriangle$ symbol indicates the AUC value of the average WSMD for multi-heads within a layer. While there are significant variations in the $\bullet$ symbols, the $\blacktriangle$ symbols are more stable. In fact, the $\blacktriangle$ symbols have a higher correlation coefficient than the $\bullet$ symbols, as shown in Table 3.

| WSMD calculation | # | $\rho \times 100$ |
|---|---|---|
| single head ($\bullet$) | 144 | 69.30 |
| multi-heads within a layer ($\blacktriangle$) | 12 | 74.50 |

Table 3: The Spearman's $\rho$ between AUC values of WSMD for $\text{PAWS}_{\text{QQP}}$ and those for $\text{PAWS}_{\text{Wiki}}$ in Fig. 4. The average WSMD for multi-heads within a layer has a higher correlation than the WSMD for a single head.

tence.

**DynaMax** (Zhelezniak et al., 2019) is a fully unsupervised and non-parametric similarity that dynamically extracts and max-pools good features depending on the sentence pair.

## 7.4 Benchmarking OT methods

The following OT methods computed from word embeddings were selected: WMD (Kusner et al., 2015) with the uniform and IDF weights. WRD (Yokoi et al., 2020) with the norm and IDF weights. SynWMD (Wei et al., 2022) with SWF weights and cost defined by cosine similarity and SWD. WMDo (Chow et al., 2019) with cost defined by cosine similarity. OPWD (Su and Hua,

2019) with cost defined by $L_2$ distance and cosine similarity. ROTS (Wang et al., 2020, 2022) using the most effective method ROTS+SWC+mean. We used Python Optimal Transport (POT) (Flamary et al., 2021) to implement simple WMD-like methods, SMD, and WSMD. We also used the OPWD part of the publicly available code for OWMD (Liu et al., 2018), but other parts could not be found.

## 7.5 Head selection to compute SAM

Which heads should we use for WSMD? Fig. 4 shows the AUC values of WMD and WSMD for each head on the $\text{PAWS}_{\text{QQP}}$ and $\text{PAWS}_{\text{Wiki}}$ dev sets. It also shows the AUC values computed for the WSMD averaged over multiple heads within a layer. Table 3 shows the Spearman's $\rho$ for AUC values in Fig. 4. These results suggest that when calculating WSMD, selecting an appropriate layer and using the average WSMD for multi-heads within that layer is likely to result in more stable performance than using a single head.

It is not desirable to change the heads or layers for each data set. Therefore, we first define *bidirectional attention variability* $\text{BV}_{\mathcal{A}}$ as a score for the layer containing SAMs that can capture the context of sentences. We then consider averaging WSMD using the SAMs of the layers selected based on $\text{BV}_{\mathcal{A}}$. For details on $\text{BV}_{\mathcal{A}}$, see Appendix E.

Fig. 5 shows the layer-wise $\text{BV}_{\mathcal{A}}$ for BERT, Sim-

| SAM$_l$ | BERT0 | | | BERT12 | | |
|---|---|---|---|---|---|---|
| | PAWS$_{QQP}$ | PAWS$_{Wiki}$ | STSB | PAWS$_{QQP}$ | PAWS$_{Wiki}$ | STSB |
| | AUC $\times$ 100 | | $\rho \times$ 100 | AUC $\times$ 100 | | $\rho \times$ 100 |
| top 1 | 9.90 | 5.22 | -0.74 | 0.26 | 0.66 | -1.98 |
| 5-12 | 6.84 | 3.55 | -0.53 | -0.04 | 0.00 | -1.32 |
| 1-12 | 7.38 | 3.32 | -0.97 | -0.06 | -0.37 | -1.34 |

Table 5: Average score improvement of WSMD for WMD-like methods with BERT.

| SAM$_l$ | SimCSE0 | | | SimCSE12 | | |
|---|---|---|---|---|---|---|
| | PAWS$_{QQP}$ | PAWS$_{Wiki}$ | STSB | PAWS$_{QQP}$ | PAWS$_{Wiki}$ | STSB |
| | AUC $\times$ 100 | | $\rho \times$ 100 | AUC $\times$ 100 | | $\rho \times$ 100 |
| top 1 | 5.44 | 1.67 | -0.62 | 0.17 | 0.59 | -0.93 |
| 5-12 | 4.39 | 0.90 | -0.30 | 0.15 | -0.06 | -1.01 |
| 1-12 | 5.79 | 1.42 | -0.73 | 0.33 | -0.03 | -1.20 |

Table 6: Average score improvement of WSMD for WMD-like methods with SimCSE.

| SAM$_l$ | RoBERTa0 | | | RoBERTa12 | | |
|---|---|---|---|---|---|---|
| | PAWS$_{QQP}$ | PAWS$_{Wiki}$ | STSB | PAWS$_{QQP}$ | PAWS$_{Wiki}$ | STSB |
| | AUC $\times$ 100 | | $\rho \times$ 100 | AUC $\times$ 100 | | $\rho \times$ 100 |
| top 1 | 11.04 | 2.49 | -0.74 | 0.01 | 0.37 | -1.37 |
| 5-12 | 11.48 | 2.43 | -0.88 | 0.13 | 0.24 | -1.38 |
| 1-12 | 10.42 | 2.28 | -1.10 | -0.11 | 0.01 | -1.41 |

Table 7: Average score improvement of WSMD for WMD-like methods with RoBERTa.

CSE, and RoBERTa on the PAWS$_{QQP}$ dev set. $BV_{\mathcal{A}}$ is high from the fifth layer in each model. Thus, we consider the following three ways to compute the average WSMD[4]: (i) The average WSMD using the best single layer from the fifth onward (top1 layer). (ii) The average WSMD over the fifth onwards (5-12 layers). (iii) The average WSMD over all the layers (1-12 layers). Table 4 presents the top1 layer chosen for each model. It is worth noting that the top1 layer selected using the PAWS$_{QQP}$ dev set was applied to all other datasets. It is also important to note that the selection of layers (i.e., 5-12 layers) in method (ii) was based only on $BV_{\mathcal{A}}$ of sentences from PAWS$_{QQP}$ dev set without any reference to the labels for the binary classification. In particular, method (iii) is fully unsupervised in terms of layer selection.

## 7.6 Results

Some WMD-like methods (WMD, WRD, Syn-WMD) are combined with SMD (as indicated as WSMD) and compared with the original method to see if an improvement by introducing WSMD. We use $\lambda = 0.5$ for WSMD. For more details about the parameters for the other methods, see Appendix D.

---

[4] We used PAWS$_{QQP}$ dev set for the layer selection because it has better performance compared to PAWS$_{Wiki}$, and also because STSB is a dataset with different properties than PAWS.

WSMD is not attempted for ROTS, OPWD and WMDo because these methods cannot be expressed in the form (4).

### 7.6.1 Comparison of WSMD Performance

Fig. 6 shows the experimental results compared to some baselines using BERT embeddings. For PAWS$_{QQP}$ at the top of Fig. 6 and PAWS$_{Wiki}$ at the middle of Fig. 6, the best-performing layer selection is the top1 layer, and improvements were observed in WSMD with WMD, WRD, and Syn-WRD for both BERT0 and BERT12. The top1 layer is chosen based on the PAWS$_{QQP}$ dev set, therefore the performance improvement is greater in PAWS$_{QQP}$ compared to PAWS$_{Wiki}$. For STSB at the bottom of Fig. 6, there is a small decrease in performance. However, especially with BERT0, performance can be improved on the PAWS datasets while still maintaining some level of performance on STSB.

Similar to Fig. 6, the experimental results for SimCSE and RoBERTa are shown in Appendix Figs. 7 and 8, respectively. We observe that the results are roughly similar to those for BERT, while the methods using RoBERTa0 show better performance compared to those using RoBERTa12 on STSB. Detailed results for each dataset are shown in Appendix Tables 19, 20, and 21.

### 7.6.2 Comparison of layer selection methods

Tables 5, 6, and 7 show the performance improvement of WSMD using different head selection. It is observed that the performance on PAWS$_{QQP}$ and PAWS$_{Wiki}$ improves for all models when the top1 layer is selected. As shown in Fig. 6, particularly significant performance improvements are observed when the 0th-layer embeddings are used.

In addition, even for unsupervised layer selection, the performance improvements are seen on PAWS$_{QQP}$ and PAWS$_{Wiki}$ for all 0th-layer embeddings. In particular, for RoBERTa on PAWS, while the performance improves when using layers from 5 to 12 based on $BV_{\mathcal{A}}$, the performance decreases when using layers from 1 to 12. This may suggest that the early layers do not have enough contextual information, as also indicated by $BV_{\mathcal{A}}$ in Fig. 5.

For STSB, although no performance improvement is observed, the performance degradation is limited to a small 2%.

### 7.7 Supplementary experiments

In addition to the abovementioned experiments, we have conducted other experiments in different settings.

To evaluate the generality of our method, we conducted experiments with DistilBERT (Sanh et al., 2019) as a model with a different number of layers than BERT, and with BERT trained with a different seed (Sellam et al., 2022). We evaluated the performance of these models on $PAWS_{QQP}$, $PAWS_{Wiki}$, and STSB. WSMD also showed strong performance on these models, especially on PAWS. More details can be found in Appendix G.

We also extended our experiments to another dataset, SICK-R (Marelli et al., 2014), using BERT to measure semantic textual similarity. Similar to STSB, the degradation in performance was within 2%. Detailed results of this additional study are also available in Appendix H.

Given the small size of the $PAWS_{QQP}$ test set, which contains only 677 sentence pairs, we also present the scores on the $PAWS_{QQP}$ train set using our method for WMD in Appendix I. The result suggests that improvements in paraphrase identification by WSMD are observed even when the dataset size increases.

These results confirm the effectiveness of our method on different datasets, using different models with SAMs.

## 8 Conclusion

Since WMD treats a sentence as a set of word embeddings and computes sentence similarity, it cannot consider the word order in the sentence. Therefore, we focused on the fact that the SAM of the input sentence obtained from the pre-trained BERT represents the relationship between words in the sentence and has information on the sentence structure. We proposed an optimal transport distance WSMD that improves existing WMD-like methods by using FGW distance that simultaneously measures the difference between word embeddings and the difference between sentence structures. We conducted experiments on paraphrase identification on PAWS and sentence similarity on STSB, confirming the proposed method boosts PAWS performance with minimal impact on STSB.

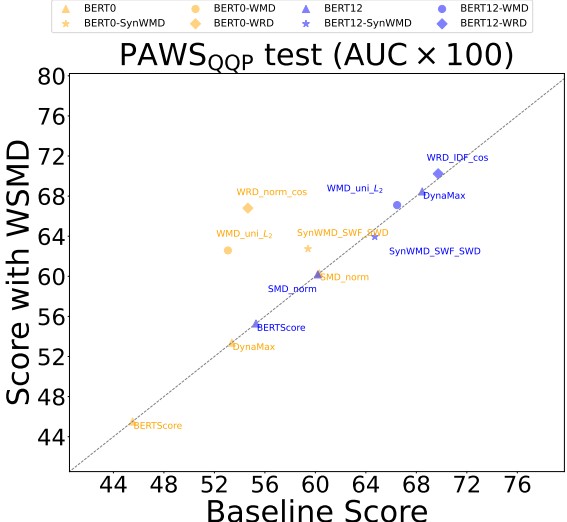

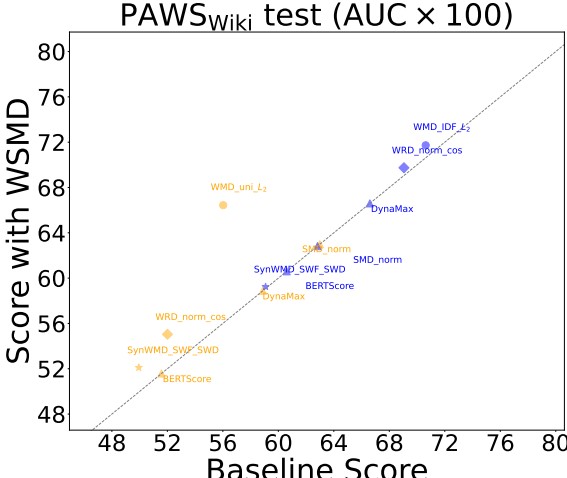

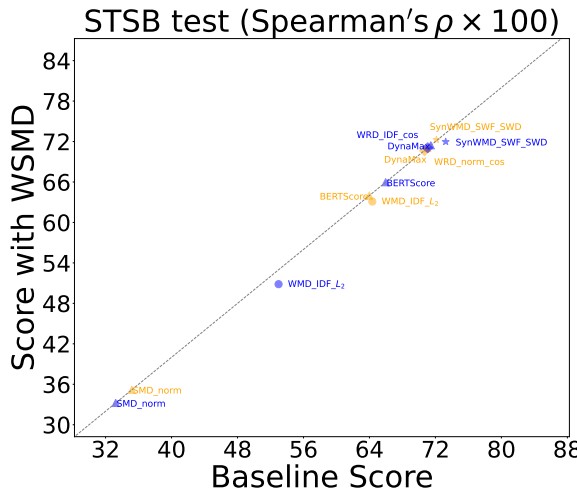

Figure 6: Performance of WSMD with several WMD-like methods (SMD, WMD, WRD, SynWMD) for BERT. The scores (AUC or Spearman's $\rho$) are compared with the original WMD-like methods. Methods that are not applicable to WSMD are positioned on the diagonal line. Values above the diagonal line represent performance improvements achieved by WSMD.

## Limitations

- As seen in Section 7.6, there was not much improvement in scores on STSB compared to PAWS. Since methods like uSIF, which consider sentences as sets of words, show good performance, it is assumed that in STSB, unlike PAWS, there are fewer sentences where word order changes significantly alter the meaning, and the SMD term of WSMD is considered noise. In fact, in STSB, it was found that there is a strong correlation between the number of common words in a sentence pair and similarity. See the Appendix J for details.

- As noted in the Section 7.6, WSMD shows a smaller performance improvement with 12th embeddings compared to 0th embeddings in PAWS. This is probably due to the fact that the 12th embeddings are contextualized, and the changes in word order within sentences in PAWS are reflected in the embeddings.

- Sentences with the same meaning do not necessarily have the same structure. For example, *I don't think it makes sense* and *it doesn't make sense* are a pair of sentences with the same meaning. Still, they have different structures, so using WSMD might decrease the sentence similarity score.

- Compared to regular WMD, as the number of SAMs used in WSMD increases, the computation time also increases. However, since WSMD can be computed independently for each SAM, parallel processing is possible if resources are available.

- WSMD cannot be applied directly to other embeddings, such as word2vec, because different tokenizers are used for each embedding. However, it may be possible to use WSMD through processing such as replacing the tokens of the model with words in the other embeddings vocabulary.

## Ethics Statement

This study complies with the ACL Ethics Policy.

## Acknowledgements

We would like to thank anonymous reviewers for their helpful advice. This study was partially supported by JSPS KAKENHI 22H05106, 22H03654, 23H03355, JST ACT-X JPMJAX200S, and JST CREST JPMJCR21N3.

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

## A   WSMD definition details

For clarity, we rewrite equation (7). By specifying the mixing ratio parameter $\lambda \in [0, 1]$, we obtain

$$\text{WSMD}((s, \mathbf{A}), (s', \mathbf{A}')) =$$

$$\min_{\mathbf{P} \in \mathbf{\Pi}(\mathbf{u}, \mathbf{u}')} \sum_{i,i'=1}^{n} \sum_{j,j'=1}^{m} \left\{ (1 - \lambda) C_{ij} \right.$$

$$\left. + \lambda k \left| A_{ii'} - A'_{jj'} \right|^2 \right\} P_{ij} P_{i'j'}$$

where $k = C_{\mathrm{M}}/A_{\mathrm{MSE}}$ is computed from

$$C_{\mathrm{M}} = \sum_{i=1}^{n} \sum_{j=1}^{m} \frac{C_{ij}}{nm}$$

$$A_{\mathrm{MSE}} = \sum_{i,i'=1}^{n} \sum_{j,j'=1}^{m} \frac{|A_{ii'} - A'_{jj'}|^2}{n^2 m^2}.$$

Note that our definition of $k$ is only to increase the interpretability of $\lambda$, and $k$ does not change the performance at all. By this definition, $C_M$ represents the average of $C_{ij}$, and $A_{MSE}$ the average of $|A_{ii'} - A'_{jj'}|^2$. From this, we can roughly consider $C_{ij} \sim C_M, |A_{ii'} - A'_{jj'}|^2 \sim A_{MSE}$. If we employ $k$, then $k|A_{ii'} - A'_{jj'}|^2 \sim C_{\mathrm{M}}/A_{\mathrm{MSE}} \cdot A_{\mathrm{MSE}} \sim C_{ij}$. With this formulation, we can use an easily interpretable value for $\lambda$, such as 0.5. By noting $\sum_{i'=1}^{n} \sum_{j'=1}^{m} P_{i'j'} = 1$, WSMD = WMD for $\lambda = 0$. For $\lambda = 1$, WSMD = $k$SMD. For an intermediate value $\lambda \in (0, 1)$, WSMD considers both WMD and SMD. We normalized $\lambda$ by introducing the factor $k$ in (7). $C_{\mathrm{M}}$ and $A_{\mathrm{MSE}}$ are interpreted as (2) and (5), respectively, by specifying $P_{ij} = u_i u'_j$ with the uniform weight (1).

## B   Decomposition of WSMD into WMD and SMD components

Let $\hat{\mathbf{P}}$ be the optimal transport matrix of (7), i.e., $\mathbf{P}$ that attains the minimum of (7). Substitute this $\hat{\mathbf{P}}$ into (2) and (5), denote them as $\text{WMD}_\lambda$ and $\text{SMD}_\lambda$, respectively. Then we rewrite equation (8) as follows:

$$\text{WSMD} = (1 - \lambda)\text{WMD}_\lambda + \lambda k \text{SMD}_\lambda.$$

In case (2) of Table 1, we computed WSMD = 8.03, $k = 688$, $\lambda = 0.5$. This is decomposed into the components $\text{WMD}_\lambda = 13.30$ and $k\text{SMD}_\lambda = 2.76$. On the other hand, we can also compute WMD = 12.55 and $k$SMD = 2.76 in the same setting, which indicates that WSMD is not a simple interpolation of WMD and $k$SMD. This is because

the optimal transport matrices in the computation of WMD, SMD, and WSMD are all different in general, and in WSMD, the optimization considers the two components simultaneously.

Interestingly, the decomposition (8) indicates that $\text{WMD}_\lambda$ can be interpreted as an improvement of WMD by utilizing $(\lambda/(1 - \lambda))k\text{SMD}_\lambda$ as a penalty term. Thus, in addition to WSMD, $\text{WMD}_\lambda$ can also be used as a sentence similarity measure, although this is not the main argument of this paper.

## C   Models

The Hugging Face models used in the experiments are presented in Table 8.

## D   Parameters for each method

Parameters are shown for each method used in the experiments. We provide explanations for terms that may be unclear or less intuitive compared to the original notation in the paper. numSVToRemove means the number of singular vectors to be removed.

### D.1   SIF

Parameters of SIF used in the experiments are shown in Table 9.

### D.2   uSIF

Parameters of uSIF used in the experiments are shown in Table 10.

### D.3   Conceptor Negation

Parameters of Conceptor Negation used in the experiments are shown in Table 11.

### D.4   ROTS

WE apply the most effective technique, ROTS+SWC+mean to ROTS. In this context, SWC+mean denotes a process that implements dimension-wise (S)caling (Ethayarajh, 2019), possesses SIF (W)eights, applies (C)onceptor Negation, and uses the *mean* in the aggregation of pair-wise cosine similarities of words. Parameters of ROTS used in the experiments are shown in Table 12. $P_{reg}$ means prior regularization, $C_{reg}$ means cosine regularization, and $E_{reg}$ means entropy regularization. $C$ also means interpolation coefficient. aggregation means how to handle different scores (*mean*, *max*, *min*, *last*, *no*).

| model | Hugging Face model |
|---|---|
| BERT | `bert-base-uncased` |
| unsup-SimCSE | `princeton-nlp/unsup-simcse-bert-base-uncased` |
| RoBERTa | `roberta-base` |
| DistilBERT | `distilbert-base-uncased` |
| BERT$_{seed0}$ | `google/multiberts-seed_0` |

Table 8: Models of Hugging Face used in the experiments.

| parameters | values |
|---|---|
| weight | SIF |
| numSVToRemove | 1 |

Table 9: Parameters of SIF used in the experiments.

| parameters | values |
|---|---|
| weight | uSIF |
| numSVToRemove | 5 |

Table 10: Parameters of uSIF used in the experiments.

## D.5 OPWD

Parameters of OPWD used in the experiments are shown in Table 13.

## D.6 WMDo

Parameter of ROTS WMDo in the experiments are shown in Table 14.

## D.7 SynWMD

Parameter of SynWMD used in the experiments are shown in Table 15.

| parameters | values |
|---|---|
| weight | SIF |
| $\alpha_{CN}$ | 2 |

Table 11: Parameters of Conceptor Negation used in the experiments.

| parameters | values |
|---|---|
| weight | SIF |
| $\alpha_{CN}$ | 2 |
| numSVToRemove | 1 |
| parser | dependency tree |
| aggregation | mean |
| normed vectors | True |
| tree depth | 5 |
| $P_{reg}$ | $[10, 10, 10, 10, 10]$ |
| $C_{reg}$ | 0 |
| $E_{reg}$ | 0 |
| $C$ | 1 |

Table 12: Parameters of ROTS used in the experiments.

| parameters | values |
|---|---|
| $\lambda_1$ | 10 |
| $\lambda_2$ | 0.03 |
| $\sigma$ | 10 |
| Iteration | 20 |

Table 13: Parameters of OPWD used in the experiments.

| parameter | value |
|---|---|
| $\delta$ | 0.2 |

Table 14: Parameter of WMDo used in the experiments.

| parameter | value |
|---|---|
| $a$ | 1 |

Table 15: Parameter of SynWMD used in the experiments.

# E  Details of attention variability

We employ *attention variability* as proposed in (Vig and Belinkov, 2019). This measures the degree of attention variability across different inputs. A high variability implies that the attention head is content-dependent, while a low variability suggests that the head is content-independent. Mathematically, attention variability $V_{\mathcal{A}}$ is defined as:

$$V_{\mathcal{A}} = \frac{\sum_{x \in X} \sum_{i=1}^{|x|} \sum_{j=1}^{i} |A_{ij}(x) - \bar{A}_{ij}|}{2 \cdot \sum_{x \in X} \sum_{i=1}^{|x|} \sum_{j=1}^{i} A_{ij}(x)} \quad (9)$$

where $\mathcal{A}$ represents the head contained in a given layer, $x$ represents sentences of length $|x|$ in the dataset $X$, $\mathcal{A}(x) = (A_{ij}(x)) \in \mathbb{R}^{|x| \times |x|}$ is the specific SAM for each sentence $x$, and $\bar{A}_{ij}$ is the average of $A_{ij}(x)$ over all $x \in X$. $V_{\mathcal{A}}$ is defined for GPT-2 (Radford et al., 2018), and the summation from 1 to $i$ for $j$ reflects the unidirectional structure of GPT-2.

We adapt the attention variability $V_{\mathcal{A}}$ to BERT by defining a *bidirectional attention variability* $BV_{\mathcal{A}}$ that measures the degree of attention variability across different inputs, where the summation for $j$ ranges from 1 to $|x|$. It is defined as follows:

$$BV_{\mathcal{A}} = \frac{\sum_{x \in X} \sum_{i=1}^{|x|} \sum_{j=1}^{|x|} |A_{ij}(x) - \bar{A}_{ij}|}{2 \cdot \sum_{x \in X} \sum_{i=1}^{|x|} \sum_{j=1}^{|x|} A_{ij}(x)}$$
$$(10)$$

# F  Detail results of experiments in Section 7

For PAWS$_{QQP}$, PAWS$_{Wiki}$, and STSB, the scores from BOWS, [CLS], uniform-weighted and IDF-weighted SMD, methods that do not require word embeddings, are shown in Tables 16, 17, and 18, respectively. The scores for methods that depend on word embeddings are shown in Tables 19, 20, and 21. Similar to Fig. 6, the experimental results for SimCSE and RoBERTa are shown in Figs. 7 and 8.

| methods | layer | weight | PAWS$_{QQP}$ (AUC $\times$ 100) | | |
| --- | --- | --- | --- | --- | --- |
| | | | BERT | SimCSE | RoBERTa |
| BOWS | | | 44.56 | 44.56 | 46.16 |
| [CLS] | | | 60.24 | 62.39 | 57.76 |
| SMD | top 1 | uniform | 59.84 | 59.70 | 60.95 |
| | 5-12 | | 58.50 | 58.04 | 59.96 |
| | 1-12 | | 56.23 | 58.35 | 57.27 |
| | top 1 | IDF | 60.25 | 59.77 | 59.48 |
| | 5-12 | | 57.99 | 59.30 | 57.19 |
| | 1-12 | | 57.82 | 57.71 | 55.62 |

Table 16: The scores on the PAWS$_{QQP}$ test set for BOWS, [CLS], uniform-weighted and IDF-weighted SMD, methods that do not require word embeddings. Note that BERT and SimCSE have identical scores for BOWS because they use the same tokenizer.

| methods | layer | weight | PAWS$_{Wiki}$ (AUC $\times$ 100) | | |
| --- | --- | --- | --- | --- | --- |
| | | | BERT | SimCSE | RoBERTa |
| BOWS | | | 48.86 | 48.86 | 50.59 |
| [CLS] | | | 57.94 | 58.02 | 56.29 |
| SMD | top 1 | uniform | 61.68 | 58.51 | 58.18 |
| | 5-12 | | 62.98 | 60.91 | 58.69 |
| | 1-12 | | 59.66 | 58.82 | 57.18 |
| | top 1 | IDF | 62.73 | 59.22 | 53.86 |
| | 5-12 | | 63.30 | 59.92 | 53.08 |
| | 1-12 | | 60.83 | 58.96 | 51.89 |

Table 17: The scores on the PAWS$_{Wiki}$ test set for BOWS, [CLS], uniform-weighted and IDF-weighted SMD, methods that do not require word embeddings. Note that BERT and SimCSE have identical scores for BOWS because they use the same tokenizer.

| methods | layer | weight | STSB (Spearman's $\rho \times$ 100) | | |
| --- | --- | --- | --- | --- | --- |
| | | | BERT | SimCSE | RoBERTa |
| BOWS | | | 68.82 | 68.82 | 67.93 |
| [CLS] | | | 41.17 | 75.23 | 40.75 |
| SMD | top 1 | uniform | 21.30 | 17.51 | 23.21 |
| | 5-12 | | 29.81 | 35.41 | 27.71 |
| | 1-12 | | 30.48 | 33.78 | 27.50 |
| | top 1 | IDF | 22.97 | 19.48 | 26.11 |
| | 5-12 | | 33.45 | 39.18 | 31.47 |
| | 1-12 | | 34.32 | 37.65 | 31.76 |

Table 18: The scores on the STSB test set for BOWS, [CLS], uniform-weighted and IDF-weighted SMD, methods that do not require word embeddings. Note that BERT and SimCSE have identical scores for BOWS because they use the same tokenizer.

| methods | layer | weight | cost | PAWS$_{QQP}$ (AUC × 100) | | | | | |
|---|---|---|---|---|---|---|---|---|---|
| | | | | BERT0 | BERT12 | SimCSE0 | SimCSE12 | RoBERTa0 | RoBERTa12 |
| Avg. Pool. | | | | 52.96 | 64.33 | 50.96 | 63.00 | 45.46 | 63.04 |
| BERTScore | | uniform | | 45.52 | 54.78 | 45.41 | 51.74 | 41.91 | 51.61 |
| | | IDF | | 45.40 | 55.29 | 45.25 | 51.98 | 42.71 | 53.01 |
| DynaMax | | | | 53.38 | 68.47 | 51.89 | 66.39 | 49.82 | 66.41 |
| SIF | | | | 53.23 | 64.15 | 51.34 | 62.83 | 45.00 | 62.76 |
| uSIF | | | | 53.22 | 64.05 | 51.47 | 62.84 | 44.75 | 62.68 |
| Con. Neg. | | | | 53.05 | 64.17 | 51.21 | 63.04 | 44.85 | 62.83 |
| ROTS | | | | 53.07 | 64.71 | 56.48 | 65.00 | 47.38 | 64.50 |
| OPWD | | uniform | $L_2$ | 53.07 | 66.48 | 53.01 | 65.58 | 48.35 | 66.69 |
| | | | cosine | 54.30 | 69.98 | 54.41 | 67.38 | 48.40 | 68.81 |
| WMDo | | uniform | cosine | 43.86 | 56.33 | 43.71 | 50.33 | 44.25 | 61.49 |
| SMD | top 1 | | | 60.31 | 60.19 | 58.40 | 58.08 | 59.97 | 58.74 |
| | 5-12 | norm | | 58.55 | 56.89 | 57.78 | 56.16 | 59.27 | 57.44 |
| | 1-12 | | | 58.35 | 56.55 | 57.55 | 56.34 | 57.39 | 56.08 |
| WMD | | uniform | $L_2$ | 53.07 | 66.48 | 53.01 | 65.58 | 48.35 | 66.69 |
| | top 1 | | | 62.58 (↑ 9.51) | 67.10 (↑ 0.62) | 57.87 (↑ 4.86) | 65.85 (↑ 0.27) | 58.96 (↑ 10.61) | 66.74 (↑ 0.05) |
| | 5-12 | | | 58.74 (↑ 5.67) | 66.77 (↑ 0.29) | 56.35 (↑ 3.34) | 65.94 (↑ 0.36) | 59.32 (↑ 10.97) | 66.95 (↑ 0.26) |
| | 1-12 | | | 59.41 (↑ 6.34) | 66.71 (↑ 0.23) | 57.54 (↑ 4.53) | 66.13 (↑ 0.55) | 58.33 (↑ 9.98) | 66.76 (↑ 0.07) |
| | | IDF | | 52.44 | 66.08 | 52.47 | 65.26 | 48.28 | 64.53 |
| | top 1 | | | 61.54 (↑ 9.10) | 66.75 (↑ 0.67) | 56.72 (↑ 4.25) | 65.58 (↑ 0.32) | 57.52 (↑ 9.24) | 64.61 (↑ 0.08) |
| | 5-12 | | | 57.90 (↑ 5.46) | 66.36 (↑ 0.28) | 55.68 (↑ 3.21) | 65.66 (↑ 0.40) | 57.78 (↑ 9.50) | 64.65 (↑ 0.12) |
| | 1-12 | | | 58.47 (↑ 6.03) | 66.31 (↑ 0.23) | 56.83 (↑ 4.36) | 65.80 (↑ 0.54) | 56.80 (↑ 8.52) | 64.39 (↓ -0.14) |
| WRD | | norm | | 54.65 | 69.07 | 53.71 | 66.99 | 48.89 | 67.45 |
| | top 1 | | | 66.80 (↑ 12.15) | 69.56 (↑ 0.49) | 60.38 (↑ 6.67) | 67.59 (↑ 0.60) | 60.87 (↑ 11.98) | 67.59 (↑ 0.14) |
| | 5-12 | | cosine | 63.84 (↑ 9.19) | 69.25 (↑ 0.18) | 58.64 (↑ 4.93) | 67.32 (↑ 0.33) | 62.34 (↑ 13.45) | 67.64 (↑ 0.19) |
| | 1-12 | | | 64.62 (↑ 9.97) | 69.36 (↑ 0.29) | 60.84 (↑ 7.13) | 67.65 (↑ 0.66) | 61.18 (↑ 12.29) | 67.54 (↑ 0.09) |
| | | IDF | | 53.94 | 69.73 | 54.01 | 67.07 | 48.88 | 66.12 |
| | top 1 | | | 67.43 (↑ 13.49) | 70.23 (↑ 0.50) | 62.09 (↑ 8.08) | 67.60 (↑ 0.53) | 60.06 (↑ 11.18) | 66.25 (↑ 0.13) |
| | 5-12 | | | 64.77 (↑ 10.83) | 69.91 (↑ 0.18) | 61.18 (↑ 7.17) | 67.40 (↑ 0.33) | 60.71 (↑ 11.83) | 66.23 (↑ 0.11) |
| | 1-12 | | | 65.47 (↑ 11.53) | 70.00 (↑ 0.27) | 63.66 (↑ 9.65) | 67.64 (↑ 0.57) | 59.79 (↑ 10.91) | 66.15 (↑ 0.03) |
| SynWMD | | SWF | cosine | 47.60 | 60.72 | 47.92 | 59.17 | 42.87 | 66.26 |
| | top 1 | | | 59.41 (↑ 11.81) | 60.73 (↑ 0.01) | 54.99 (↑ 7.07) | 58.89 (↓ -0.28) | 59.73 (↑ 16.86) | 66.10 (↓ -0.16) |
| | 5-12 | | | 56.27 (↑ 8.67) | 60.37 (↓ -0.35) | 54.78 (↑ 6.86) | 59.03 (↓ -0.14) | 60.43 (↑ 17.56) | 66.36 (↑ 0.10) |
| | 1-12 | | | 56.57 (↑ 8.97) | 60.26 (↓ -0.46) | 55.82 (↑ 7.90) | 59.17 (–) | 58.60 (↑ 15.73) | 65.80 (↓ -0.46) |
| | | | SWD | 59.41 | 64.71 | 58.47 | 62.47 | 57.85 | 70.14 |
| | top 1 | | | 62.74 (↑ 3.33) | 63.95 (↓ -0.76) | 60.15 (↑ 1.68) | 62.03 (↓ -0.44) | 64.20 (↑ 6.35) | 69.97 (↓ -0.17) |
| | 5-12 | | | 60.63 (↑ 1.22) | 63.88 (↓ -0.83) | 59.27 (↑ 0.80) | 62.10 (↓ -0.37) | 63.41 (↑ 5.56) | 70.11 (↓ -0.03) |
| | 1-12 | | | 60.86 (↑ 1.45) | 63.78 (↓ -0.93) | 59.65 (↑ 1.18) | 62.09 (↓ -0.38) | 62.94 (↑ 5.09) | 69.87 (↓ -0.27) |

Table 19: The scores on the PAWS$_{QQP}$ test set for the methods using the embeddings from BERT, SimCSE, and RoBERTa. For WMD, WRD, and SynWMD, the scores of WSMD and the original method are compared for each layer selection. Score increases are highlighted in red and score decreases are highlighted in blue. Maximum scores for each embedding are underlined.

| methods | layer | weight | cost | PAWS$_{\text{Wiki}}$ (AUC × 100) | | | | | |
| | | | | BERT0 | BERT12 | SimCSE0 | SimCSE12 | RoBERTa0 | RoBERTa12 |
|---|---|---|---|---|---|---|---|---|---|
| Avg. Pool. | | | | 49.43 | 60.06 | 49.68 | 56.71 | 51.11 | 59.13 |
| BERTScore | | uniform | | 51.58 | 60.60 | 52.66 | 57.02 | 52.99 | 57.57 |
| | | IDF | | 51.58 | 60.60 | 52.73 | 56.92 | 52.03 | 55.79 |
| DynaMax | | | | 58.88 | 66.60 | 51.99 | 62.37 | 53.87 | 65.05 |
| SIF | | | | 49.31 | 59.90 | 49.62 | 56.70 | 51.09 | 59.23 |
| uSIF | | | | 49.29 | 60.03 | 49.59 | 56.75 | 51.13 | 59.27 |
| Con. Neg. | | | | 49.40 | 60.31 | 49.65 | 56.78 | 51.18 | 59.24 |
| ROTS | | | | 49.85 | 60.68 | 53.74 | 57.45 | 52.16 | 60.48 |
| OPWD | | uniform | $L_2$ | 56.02 | 70.40 | 56.00 | 64.78 | 55.36 | 65.61 |
| | | | cosine | 52.00 | 68.87 | 52.08 | 61.87 | 52.51 | 64.38 |
| WMDo | | uniform | cosine | 48.48 | 58.56 | 48.33 | 54.41 | 41.28 | 51.79 |
| SMD | top 1 | | | 61.68 | 61.53 | 59.28 | 57.94 | 58.42 | 58.32 |
| | 5-12 | norm | | 62.99 | 62.82 | 60.95 | 60.82 | 59.07 | 59.32 |
| | 1-12 | | | 60.17 | 59.93 | 59.79 | 58.91 | 57.47 | 58.24 |
| WMD | | uniform | | 56.02 | 70.40 | 56.00 | 64.78 | 55.36 | 65.61 |
| | top 1 | | | 66.44 (↑ 10.42) | 71.36 (↑ 0.96) | 58.23 (↑ 2.23) | 65.51 (↑ 0.73) | 59.58 (↑ 4.22) | 65.91 (↑ 0.30) |
| | 5-12 | | | 63.43 (↑ 7.41) | 70.51 (↑ 0.11) | 57.12 (↑ 1.12) | 64.80 (↑ 0.02) | 59.43 (↑ 4.07) | 65.81 (↑ 0.20) |
| | 1-12 | | $L_2$ | 62.26 (↑ 6.24) | 69.88 (↓ -0.52) | 57.78 (↑ 1.78) | 64.84 (↑ 0.06) | 59.09 (↑ 3.73) | 65.62 (↑ 0.01) |
| | | IDF | | 55.44 | 70.62 | 55.42 | 64.48 | 50.96 | 60.24 |
| | top 1 | | | 65.63 (↑ 10.19) | 71.72 (↑ 1.10) | 57.45 (↑ 2.03) | 65.26 (↑ 0.78) | 53.80 (↑ 2.84) | 60.33 (↑ 0.09) |
| | 5-12 | | | 62.36 (↑ 6.92) | 70.72 (↑ 0.10) | 56.37 (↑ 0.95) | 64.37 (↓ -0.11) | 53.40 (↑ 2.44) | 59.93 (↓ -0.31) |
| | 1-12 | | | 61.24 (↑ 5.80) | 70.03 (↓ -0.59) | 56.98 (↑ 1.56) | 64.40 (↓ -0.08) | 52.94 (↑ 1.98) | 59.51 (↓ -0.73) |
| WRD | | norm | | 52.01 | 69.06 | 52.01 | 62.00 | 52.44 | 64.75 |
| | top 1 | | | 55.04 (↑ 3.03) | 69.75 (↑ 0.69) | 53.45 (↑ 1.44) | 62.62 (↑ 0.62) | 54.04 (↑ 1.60) | 65.05 (↑ 0.30) |
| | 5-12 | | | 54.03 (↑ 2.02) | 69.26 (↑ 0.20) | 52.90 (↑ 0.89) | 62.17 (↑ 0.17) | 54.12 (↑ 1.68) | 65.04 (↑ 0.29) |
| | 1-12 | | cosine | 54.35 (↑ 2.34) | 69.22 (↑ 0.16) | 53.50 (↑ 1.49) | 62.37 (↑ 0.37) | 54.14 (↑ 1.70) | 65.01 (↑ 0.26) |
| | | IDF | | 51.60 | 68.72 | 51.79 | 61.08 | 49.34 | 58.30 |
| | top 1 | | | 54.47 (↑ 2.87) | 69.43 (↑ 0.71) | 53.37 (↑ 1.58) | 61.71 (↑ 0.63) | 50.35 (↑ 1.01) | 58.59 (↑ 0.29) |
| | 5-12 | | | 53.59 (↑ 1.99) | 68.92 (↑ 0.20) | 52.83 (↑ 1.04) | 61.21 (↑ 0.13) | 50.34 (↑ 1.00) | 58.44 (↑ 0.14) |
| | 1-12 | | | 53.90 (↑ 2.30) | 68.86 (↑ 0.14) | 53.49 (↑ 1.70) | 61.39 (↑ 0.31) | 50.32 (↑ 0.98) | 58.31 (↑ 0.01) |
| SynWMD | | | cosine | 48.55 | 59.01 | 48.80 | 55.19 | 51.63 | 64.98 |
| | top 1 | | | 51.19 (↑ 2.64) | 59.32 (↑ 0.31) | 50.47 (↑ 1.67) | 55.73 (↑ 0.54) | 54.40 (↑ 2.77) | 65.80 (↑ 0.82) |
| | 5-12 | | | 50.24 (↑ 1.69) | 58.75 (↓ -0.26) | 49.75 (↑ 0.95) | 55.02 (↓ -0.17) | 54.47 (↑ 2.84) | 65.74 (↑ 0.76) |
| | 1-12 | SWF | | 50.45 (↑ 1.90) | 58.35 (↓ -0.66) | 50.20 (↑ 1.40) | 54.97 (↓ -0.22) | 54.48 (↑ 2.85) | 65.49 (↑ 0.51) |
| | | | SWD | 49.94 | 59.08 | 51.36 | 56.54 | 54.40 | 67.23 |
| | top 1 | | | 52.11 (↑ 2.17) | 59.24 (↑ 0.16) | 52.43 (↑ 1.07) | 56.75 (↑ 0.21) | 56.93 (↑ 2.53) | 67.65 (↑ 0.42) |
| | 5-12 | | | 51.22 (↑ 1.28) | 58.73 (↓ -0.35) | 51.78 (↑ 0.42) | 56.11 (↓ -0.43) | 56.93 (↑ 2.53) | 67.55 (↑ 0.32) |
| | 1-12 | | | 51.31 (↑ 1.37) | 58.31 (↓ -0.77) | 51.95 (↑ 0.59) | 55.94 (↓ -0.60) | 56.84 (↑ 2.44) | 67.25 (↑ 0.02) |

Table 20: The scores on the PAWS$_{\text{Wiki}}$ test set for the methods using the embeddings from BERT, SimCSE, and RoBERTa. For WMD, WRD, and SynWMD, the scores of WSMD and the original method are compared for each layer selection. Score increases are highlighted in red and score decreases are highlighted in blue. Maximum scores for each embedding are underlined.

| methods | layer | weight | cost | STSB (Spearman's $\rho \times 100$) | | | | | |
|---|---|---|---|---|---|---|---|---|---|
| | | | | BERT0 | BERT12 | SimCSE0 | SimCSE12 | RoBERTa0 | RoBERTa12 |
| Avg. Pool. | | | | 69.16 | 69.07 | 60.83 | 77.54 | 70.25 | 64.09 |
| BERTScore | | uniform | | 63.27 | 65.13 | 62.90 | 70.59 | 63.38 | 58.71 |
| | | IDF | | 63.99 | 65.96 | 63.72 | 70.96 | 64.09 | 59.82 |
| DynaMax | | | | 71.02 | 71.44 | 68.36 | 77.67 | 70.27 | 66.16 |
| SIF | | | | 68.28 | 69.12 | 57.66 | 76.61 | 68.70 | 64.82 |
| uSIF | | | | 68.02 | 68.98 | 57.80 | 76.55 | 69.35 | 64.93 |
| Con. Neg. | | | | 68.35 | 69.09 | 60.65 | 77.19 | 69.68 | 64.56 |
| ROTS | | | | 68.69 | 69.27 | 61.86 | 77.26 | 69.58 | 64.94 |
| OPWD | | uniform | $L_2$ | 60.78 | 49.62 | 60.73 | 64.88 | 58.79 | 44.43 |
| | | | cosine | 68.19 | 69.78 | 63.76 | 77.00 | 67.25 | 63.98 |
| WMDo | | uniform | cosine | 67.61 | 68.74 | 62.17 | 76.18 | 64.63 | 61.16 |
| SMD | top 1 | norm | | 23.71 | 22.43 | 18.28 | 17.72 | 27.00 | 24.15 |
| | 5-12 | | | 34.21 | 32.22 | 38.64 | 37.11 | 32.29 | 30.11 |
| | 1-12 | | | 35.18 | 33.23 | 37.00 | 35.57 | 32.54 | 29.69 |
| WMD | | | | 60.78 | 49.62 | 60.73 | 64.88 | 58.79 | 44.43 |
| | top 1 | uniform | | 58.50 (↓ -2.28) | 46.25 (↓ -3.37) | 59.23 (↓ -1.50) | 63.17 (↓ -1.71) | 57.23 (↓ -1.56) | 42.48 (↓ -1.95) |
| | 5-12 | | | 58.96 (↓ -1.82) | 47.16 (↓ -2.46) | 59.56 (↓ -1.17) | 62.81 (↓ -2.07) | 56.89 (↓ -1.90) | 42.25 (↓ -2.18) |
| | 1-12 | | $L_2$ | 58.23 (↓ -2.55) | 46.97 (↓ -2.65) | 58.99 (↓ -1.74) | 62.25 (↓ -2.63) | 56.52 (↓ -2.27) | 42.07 (↓ -2.36) |
| | | | | 64.35 | 52.98 | 64.30 | 66.27 | 62.78 | 46.97 |
| | top 1 | IDF | | 62.72 (↓ -1.63) | 49.86 (↓ -3.12) | 63.26 (↓ -1.04) | 64.74 (↓ -1.53) | 61.49 (↓ -1.29) | 45.04 (↓ -1.93) |
| | 5-12 | | | 63.10 (↓ -1.25) | 50.77 (↓ -2.21) | 63.71 (↓ -0.59) | 64.68 (↓ -1.59) | 61.27 (↓ -1.51) | 45.00 (↓ -1.97) |
| | 1-12 | | | 62.56 (↓ -1.79) | 50.85 (↓ -2.13) | 63.27 (↓ -1.03) | 64.40 (↓ -1.87) | 61.09 (↓ -1.69) | 45.05 (↓ -1.92) |
| WRD | | | | 70.68 | 70.30 | 65.96 | 76.87 | 69.82 | 64.09 |
| | top 1 | norm | | 70.60 (↓ -0.08) | 70.02 (↓ -0.28) | 65.58 (↓ -0.38) | 76.83 (↓ -0.04) | 69.75 (↓ -0.07) | 63.63 (↓ -0.46) |
| | 5-12 | | | 70.65 (↓ -0.03) | 70.15 (↓ -0.15) | 65.65 (↓ -0.31) | 76.83 (↓ -0.04) | 69.71 (↓ -0.11) | 63.58 (↓ -0.51) |
| | 1-12 | | cosine | 70.57 (↓ -0.11) | 70.15 (↓ -0.15) | 65.40 (↓ -0.56) | 76.82 (↓ -0.05) | 69.71 (↓ -0.11) | 63.56 (↓ -0.53) |
| | | | | 70.55 | 71.06 | 67.01 | 77.07 | 69.81 | 65.91 |
| | top 1 | IDF | | 70.39 (↓ -0.16) | 70.92 (↓ -0.14) | 66.69 (↓ -0.32) | 77.06 (↓ -0.01) | 69.64 (↓ -0.17) | 65.50 (↓ -0.41) |
| | 5-12 | | | 70.46 (↓ -0.09) | 71.00 (↓ -0.06) | 66.76 (↓ -0.25) | 77.11 (↑ 0.04) | 69.60 (↓ -0.21) | 65.47 (↓ -0.44) |
| | 1-12 | | | 70.33 (↓ -0.22) | 71.08 (↑ 0.02) | 66.58 (↓ -0.43) | 77.15 (↑ 0.08) | 69.58 (↓ -0.23) | 65.49 (↓ -0.42) |
| SynWMD | | | | 72.06 | 72.60 | 68.00 | 78.71 | 71.69 | 68.95 |
| | top 1 | | cosine | 71.74 (↓ -0.32) | 69.84 (↓ -2.76) | 67.61 (↓ -0.39) | 77.38 (↓ -1.33) | 70.88 (↓ -0.81) | 67.06 (↓ -1.89) |
| | 5-12 | | | 71.94 (↓ -0.12) | 70.89 (↓ -1.71) | 68.05 (↑ 0.05) | 77.30 (↓ -1.41) | 70.74 (↓ -0.95) | 67.19 (↓ -1.76) |
| | 1-12 | SWF | | 71.26 (↓ -0.80) | 70.73 (↓ -1.87) | 67.42 (↓ -0.58) | 77.05 (↓ -1.66) | 70.30 (↓ -1.39) | 67.07 (↓ -1.88) |
| | | | | 72.12 | 73.25 | 67.01 | 79.15 | 72.68 | 70.34 |
| | top 1 | | SWD | 72.13 (↑ 0.01) | 71.05 (↓ -2.20) | 66.93 (↓ -0.08) | 78.19 (↓ -0.96) | 72.14 (↓ -0.54) | 68.74 (↓ -1.60) |
| | 5-12 | | | 72.27 (↑ 0.15) | 71.93 (↓ -1.32) | 67.45 (↑ 0.44) | 78.15 (↓ -1.00) | 72.11 (↓ -0.57) | 68.94 (↓ -1.40) |
| | 1-12 | | | 71.73 (↓ -0.39) | 71.97 (↓ -1.28) | 66.96 (↓ -0.05) | 78.05 (↓ -1.10) | 71.79 (↓ -0.89) | 68.98 (↓ -1.36) |

Table 21: The scores on the STSB test set for the methods using the embeddings from BERT, SimCSE, and RoBERTa. For WMD, WRD, and SynWMD, the scores of WSMD and the original method are compared for each layer selection. Score increases are highlighted in red and score decreases are highlighted in blue. Maximum scores for each embedding are underlined.

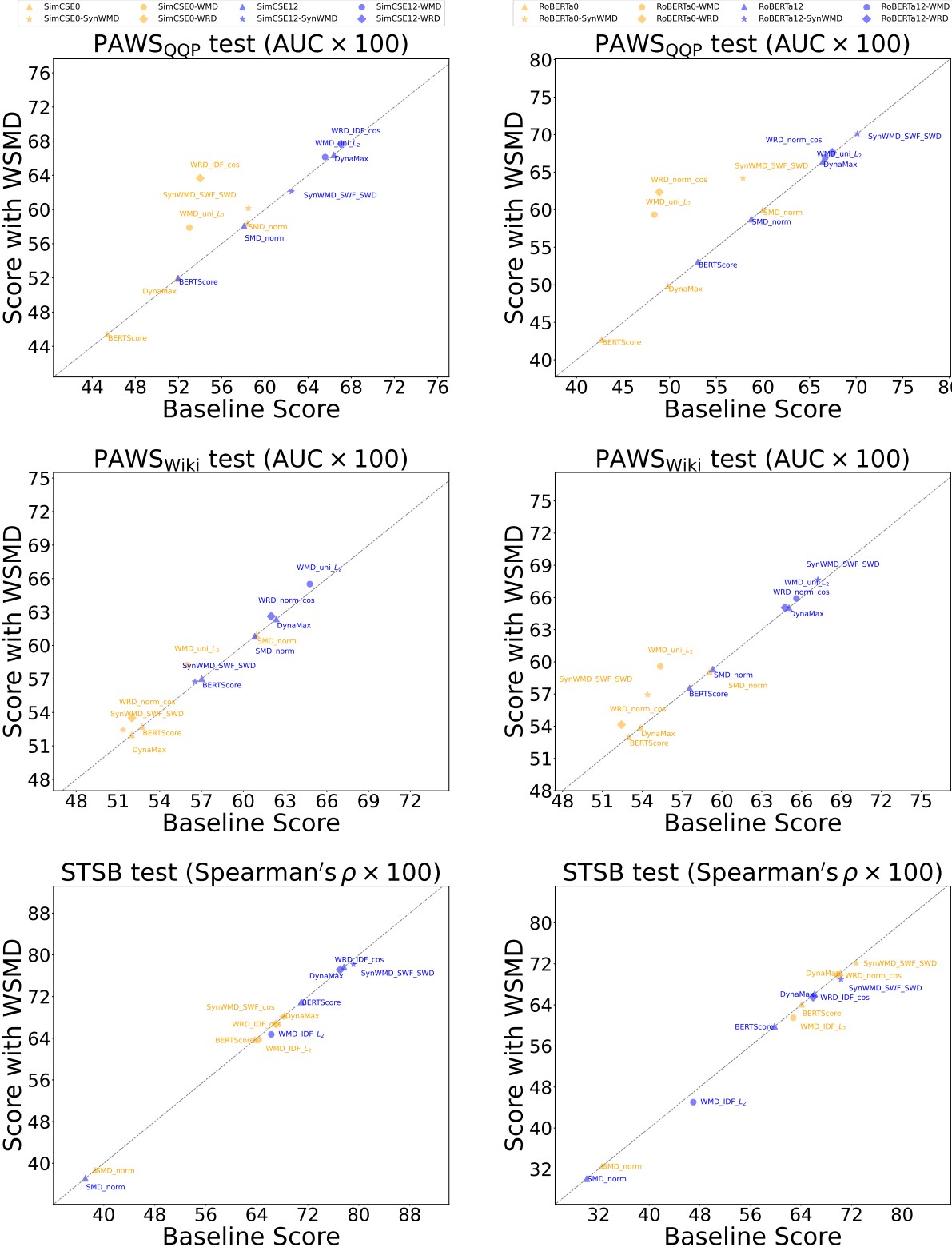

Figure 7: Performance of WSMD with several WMD-like methods (SMD, WMD, WRD, SynWMD) for SimCSE. The scores (AUC or Spearman's $\rho$) are compared with the original WMD-like methods. Methods that are not applicable to WSMD are positioned on the diagonal line. Values above the diagonal line represent performance improvements achieved by WSMD.

Figure 8: Performance of WSMD with several WMD-like methods (SMD, WMD, WRD, SynWMD) for RoBERTa. The scores (AUC or Spearman's $\rho$) are compared with the original WMD-like methods. Methods that are not applicable to WSMD are positioned on the diagonal line. Values above the diagonal line represent performance improvements achieved by WSMD.

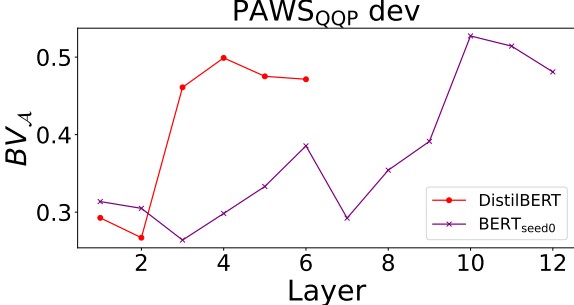

Figure 9: The layer-wise $BV_{\mathcal{A}}$ for DistilBERT and $BERT_{seed0}$ on the $PAWS_{QQP}$ dev set. $BV_{\mathcal{A}}$ is high from the third layer in DistilBERT and from the fifth layer in $BERT_{seed0}$.

|  | DistilBERT | $BERT_{seed0}$ |
| --- | --- | --- |
| 0th Embs. | 4 | 9 |
| 6th Embs. | 4 | – |
| 12th Embs. | – | 9 |

Table 22: Top1 layer for DistilBERT and $BERT_{seed0}$. The layer was selected from the third onward for DistilBERT and the fifth onward for $BERT_{seed0}$ based on the AUC of the average WSMD for a single layer using the $PAWS_{QQP}$ dev set.

## G Experiments with other models

To evaluate the generality of our method, we conducted experiments not only with BERT, SimCSE, and RoBERTa, but also with DistilBERT (Sanh et al., 2019) and BERT trained with a different seed (Sellam et al., 2022). DistilBERT is a six-layer model distilled from BERT using BERT's weights as initialization. Since BERT has 12 layers, this initialization selects one layer from each corresponding pair of layers. For the model trained with a different seed, we chose BERT with seed 0, referred to as $BERT_{seed0}$.

We used DistilBERT and $BERT_{seed0}$ from the Hugging Face transformers library (Wolf et al., 2020) and see Table 8 in Appendix C for details.

Similar to Fig. 5, Fig. 9 shows the plots of $BV_{\mathcal{A}}$ for each layer of DistilBERT and $BERT_{seed0}$ on the $PAWS_{QQP}$ dev set. For DistilBERT, $BV_{\mathcal{A}}$ is high from the third layer, while for $BERT_{seed0}$ it is high from the fifth layer, although unlike BERT, the value decreases at the seventh layer.

Therefore, for DistilBERT, we compare performance using WMD on the $PAWS_{QQP}$ dev set with three layer selection methods: the best single layer

from the third (top1 layer), third to sixth (3-6 layers), and all layers (1-6 layers). For $BERT_{seed0}$, we compare performance using the same three settings as for BERT on the $PAWS_{QQP}$ dev set.

Table 22 shows the top1 layer for each model.

The scores for methods that depend on word embeddings are shown in Tables 23, 24, and 25.

## H Experiments with SICK-R

In addition to STSB, we also extended our experiments to another dataset, SICK-R (Marelli et al., 2014), using BERT to measure semantic textual similarity.

Similar to STSB, SICK-R contains human-annotated scores for English sentence pairs, reflecting the average similarity on a six-point scale.

Table 26 shows the number of sentence pairs in the SICK-R test set.

The scores for methods that depend on word embeddings are shown in Table 27.

## I Experiments with $PAWS_{QQP}$ train set

As seen in Section 7.7, the size of the $PAWS_{QQP}$ test set is small and the dataset contains only 677 sentence pairs. To alleviate this problem, we present the scores on the $PAWS_{QQP}$ train set using our method.

Table 28 shows the number of sentence pairs in the $PAWS_{QQP}$ train set.

The scores for methods that depend on word embeddings are shown in Table 29.

It should be noted that the top layer was determined based on the first 1500 sentence pairs, so it implies a potential bias in the layer selection.

## J Correlation between common words count and gold score in sentence pairs.

For $PAWS_{QQP}$, $PAWS_{Wiki}$, and STSB, We present scatter plots of the number of common words in sentence pairs against their respective gold scores in Fig. 10. Compared to PAWS, STSB shows a strong correlation between the number of common words and the gold score. Therefore, in STSB, there is no need to worry about a gold score decrease for sentence pairs with high word overlap. This suggests that even when sentences are considered as sets of words, good performance can be achieved.

| methods | layer | weight | cost | PAWS$_{QQP}$ (AUC × 100) | | | |
|---|---|---|---|---|---|---|---|
| | | | | DistilBERT0 | DistilBERT12 | BERT$_{seed0}$0 | BERT$_{seed0}$12 |
| Avg. Pool. | | | | 52.02 | 63.22 | 52.80 | 63.94 |
| BERTScore | | uniform | | 45.58 | 54.74 | 45.50 | 52.05 |
| | | IDF | | 45.46 | 55.30 | 45.41 | 52.46 |
| DynaMax | | | | 52.02 | 70.00 | 54.59 | 66.71 |
| SIF | | | | 52.26 | 63.03 | 52.79 | 63.68 |
| uSIF | | | | 52.29 | 63.06 | 52.85 | 63.60 |
| Con. Neg. | | | | 52.15 | 63.12 | 52.64 | 63.76 |
| ROTS | | | | 51.71 | 64.95 | 55.25 | 64.37 |
| OPWD | | uniform | $L_2$ | 53.03 | 67.33 | 53.53 | 66.99 |
| | | | cosine | 54.73 | 70.93 | 55.69 | 68.43 |
| WMDo | | uniform | cosine | 44.18 | 55.42 | 44.07 | 58.53 |
| WMD | | uniform | $L_2$ | 53.03 | 67.33 | 53.53 | 66.99 |
| | top 1 | | | 59.47 (↑ 6.44) | 66.99 (↓ -0.34) | 71.95 (↑ 18.42) | 67.24 (↑ 0.25) |
| | 5-12 | | | 58.87 (↑ 5.84) | 67.52 (↑ 0.19) | 66.64 (↑ 13.11) | 67.45 (↑ 0.46) |
| | 1-12 | | | 59.01 (↑ 5.98) | 67.38 (↑ 0.05) | 65.25 (↑ 11.72) | 67.59 (↑ 0.60) |
| | | IDF | | 52.44 | 66.79 | 53.06 | 66.69 |
| | top 1 | | | 58.41 (↑ 5.97) | 66.33 (↓ -0.46) | 70.80 (↑ 17.74) | 67.35 (↑ 0.66) |
| | 5-12 | | | 57.97 (↑ 5.53) | 66.98 (↑ 0.19) | 65.63 (↑ 12.57) | 67.23 (↑ 0.54) |
| | 1-12 | | | 58.07 (↑ 5.63) | 66.87 (↑ 0.08) | 64.18 (↑ 11.12) | 67.29 (↑ 0.60) |
| WRD | | norm | cosine | 53.25 | 70.39 | 56.00 | 67.56 |
| | top 1 | | | 63.02 (↑ 9.77) | 70.35 (↓ -0.04) | 71.65 (↑ 15.65) | 67.93 (↑ 0.37) |
| | 5-12 | | | 62.88 (↑ 9.63) | 70.51 (↑ 0.12) | 68.56 (↑ 12.56) | 67.79 (↑ 0.23) |
| | 1-12 | | | 62.95 (↑ 9.70) | 70.50 (↑ 0.11) | 68.27 (↑ 12.27) | 68.00 (↑ 0.44) |
| | | IDF | | 54.46 | 70.59 | 55.49 | 68.32 |
| | top 1 | | | 64.37 (↑ 9.91) | 70.46 (↓ -0.13) | 71.83 (↑ 16.34) | 68.63 (↑ 0.31) |
| | 5-12 | | | 65.08 (↑ 10.62) | 70.74 (↑ 0.15) | 68.85 (↑ 13.36) | 68.49 (↑ 0.17) |
| | 1-12 | | | 64.97 (↑ 10.51) | 70.71 (↑ 0.12) | 68.71 (↑ 13.22) | 68.67 (↑ 0.35) |
| SynWMD | | SWF | cosine | 47.43 | 60.98 | 47.47 | 60.52 |
| | top 1 | | | 54.58 (↑ 7.15) | 59.40 (↓ -1.58) | 64.53 (↑ 17.06) | 60.29 (↓ -0.23) |
| | 5-12 | | | 56.21 (↑ 8.78) | 60.47 (↓ -0.51) | 60.43 (↑ 12.96) | 60.17 (↓ -0.35) |
| | 1-12 | | | 55.76 (↑ 8.33) | 60.17 (↓ -0.81) | 59.66 (↑ 12.19) | 60.27 (↓ -0.25) |
| | | | SWD | 58.43 | 64.47 | 59.51 | 65.29 |
| | top 1 | | | 59.73 (↑ 1.30) | 62.80 (↓ -1.67) | 66.72 (↑ 7.21) | 63.38 (↓ -1.91) |
| | 5-12 | | | 60.09 (↑ 1.66) | 63.73 (↓ -0.74) | 63.84 (↑ 4.33) | 63.94 (↓ -1.35) |
| | 1-12 | | | 60.06 (↑ 1.63) | 63.50 (↓ -0.97) | 63.08 (↑ 3.57) | 64.18 (↓ -1.11) |

Table 23: The scores on the PAWS$_{QQP}$ test set for the methods using the embeddings from DistilBERT and BERT$_{seed0}$. For WMD, WRD, and SynWMD, the scores of WSMD and the original method are compared for each layer selection. Score increases are highlighted in red and score decreases are highlighted in blue. Maximum scores for each embedding are underlined.

| methods | layer | weight | cost | PAWS$_{Wiki}$ (AUC × 100) | | | |
|---|---|---|---|---|---|---|---|
| | | | | DistilBERT0 | DistilBERT12 | BERT$_{seed0}$0 | BERT$_{seed0}$12 |
| Avg. Pool. | | | | 50.17 | 57.25 | 49.93 | 60.22 |
| BERTScore | | uniform | | 52.58 | 56.08 | 52.60 | 57.21 |
| | | IDF | | 52.73 | 56.12 | 52.68 | 57.32 |
| DynaMax | | | | 52.38 | 59.34 | 52.50 | 56.44 |
| SIF | | | | 50.10 | 57.09 | 49.89 | 60.00 |
| uSIF | | | | 50.05 | 57.20 | 49.86 | 60.16 |
| Con. Neg. | | | | 50.11 | 57.44 | 49.89 | 60.55 |
| ROTS | | | | 54.93 | 57.68 | 54.76 | 60.62 |
| OPWD | | uniform | $L_2$ | 55.78 | 69.95 | 55.69 | 70.90 |
| | | | cosine | 51.98 | 66.96 | 52.33 | 68.76 |
| WMDo | | uniform | cosine | 48.45 | 57.16 | 48.62 | 58.79 |
| WMD | | uniform | $L_2$ | 55.78 | 69.95 | 55.69 | 70.90 |
| | top 1 | | | 58.64 (↑ 2.86) | 64.85 (↓ -5.10) | 68.26 (↑ 12.57) | 68.56 (↓ -2.34) |
| | 5-12 | | | 58.15 (↑ 2.37) | 69.20 (↓ -0.75) | 64.10 (↑ 8.41) | 69.16 (↓ -1.74) |
| | 1-12 | | | 58.63 (↑ 2.85) | 68.79 (↓ -1.16) | 62.11 (↑ 6.42) | 68.30 (↓ -2.60) |
| | | IDF | | 55.20 | 70.19 | 55.18 | 71.15 |
| | top 1 | | | 57.58 (↑ 2.38) | 64.36 (↓ -5.83) | 67.10 (↑ 11.92) | 69.42 (↓ -1.73) |
| | 5-12 | | | 57.36 (↑ 2.16) | 69.36 (↓ -0.83) | 63.00 (↑ 7.82) | 69.80 (↓ -1.35) |
| | 1-12 | | | 57.80 (↑ 2.60) | 68.93 (↓ -1.26) | 61.06 (↑ 5.88) | 68.63 (↓ -2.52) |
| WRD | | norm | cosine | 52.21 | 67.07 | 52.45 | 68.79 |
| | top 1 | | | 54.75 (↑ 2.54) | 66.83 (↓ -0.24) | 60.15 (↑ 7.70) | 69.50 (↑ 0.71) |
| | 5-12 | | | 53.98 (↑ 1.77) | 67.28 (↑ 0.21) | 57.44 (↑ 4.99) | 69.16 (↑ 0.37) |
| | 1-12 | | | 54.34 (↑ 2.13) | 67.32 (↑ 0.25) | 56.62 (↑ 4.17) | 68.99 (↑ 0.20) |
| | | IDF | | 51.76 | 66.69 | 52.08 | 68.72 |
| | top 1 | | | 54.39 (↑ 2.63) | 66.28 (↓ -0.41) | 59.70 (↑ 7.62) | 69.55 (↑ 0.83) |
| | 5-12 | | | 53.73 (↑ 1.97) | 66.89 (↑ 0.20) | 57.15 (↑ 5.07) | 69.15 (↑ 0.43) |
| | 1-12 | | | 54.11 (↑ 2.35) | 66.92 (↑ 0.23) | 56.38 (↑ 4.30) | 68.93 (↑ 0.21) |
| SynWMD | | SWF | cosine | 48.90 | 57.08 | 48.69 | 59.52 |
| | top 1 | | | 51.00 (↑ 2.10) | 55.20 (↓ -1.88) | 56.64 (↑ 7.95) | 59.07 (↓ -0.45) |
| | 5-12 | | | 50.81 (↑ 1.91) | 56.83 (↓ -0.25) | 53.93 (↑ 5.24) | 58.82 (↓ -0.70) |
| | 1-12 | | | 51.09 (↑ 2.19) | 56.70 (↓ -0.38) | 52.85 (↑ 4.16) | 58.47 (↓ -1.05) |
| | | | SWD | 52.28 | 57.10 | 52.06 | 59.11 |
| | top 1 | | | 52.45 (↑ 0.17) | 55.09 (↓ -2.01) | 57.08 (↑ 5.02) | 58.64 (↓ -0.47) |
| | 5-12 | | | 53.14 (↑ 0.86) | 56.73 (↓ -0.37) | 55.11 (↑ 3.05) | 58.40 (↓ -0.71) |
| | 1-12 | | | 53.20 (↑ 0.92) | 56.58 (↓ -0.52) | 54.18 (↑ 2.12) | 58.06 (↓ -1.05) |

Table 24: The scores on the PAWS$_{Wiki}$ test set for the methods using the embeddings from DistilBERT and BERT$_{seed0}$. For WMD, WRD, and SynWMD, the scores of WSMD and the original method are compared for each layer selection. Score increases are highlighted in red and score decreases are highlighted in blue. Maximum scores for each embedding are underlined.

| methods | layer | weight | cost | STSB (Spearman's $\rho \times 100$) | | | |
|---|---|---|---|---|---|---|---|
| | | | | DistilBERT0 | DistilBERT12 | $\text{BERT}_{\text{seed0}}0$ | $\text{BERT}_{\text{seed0}}12$ |
| Avg. Pool. | | | | 66.33 | 71.83 | 61.98 | 51.78 |
| BERTScore | | uniform | | 63.30 | 63.01 | 63.31 | 60.15 |
| | | IDF | | 64.16 | 63.92 | 64.22 | 60.97 |
| DynaMax | | | | 70.61 | 72.66 | 69.35 | 51.05 |
| SIF | | | | 65.35 | 71.76 | 59.20 | 47.38 |
| uSIF | | | | 65.25 | 71.64 | 59.34 | 47.40 |
| Con. Neg. | | | | 65.81 | 71.86 | 61.76 | 52.44 |
| ROTS | | | | 66.61 | 71.98 | 62.94 | 52.90 |
| OPWD | | uniform | $L_2$ | 60.59 | 52.63 | 61.05 | 42.70 |
| | | | cosine | 67.29 | 70.93 | 64.74 | 48.08 |
| WMDo | | uniform | cosine | 66.59 | 70.13 | 63.48 | 42.52 |
| WMD | | uniform | $L_2$ | 60.59 | 52.63 | 61.05 | 42.70 |
| | top 1 | | | 52.78 (↓ -7.81) | 44.94 (↓ -7.69) | 51.13 (↓ -9.92) | 34.24 (↓ -8.46) |
| | 5-12 | | | 58.50 (↓ -2.09) | 50.04 (↓ -2.59) | 56.31 (↓ -4.74) | 40.36 (↓ -2.34) |
| | 1-12 | | | 58.07 (↓ -2.52) | 49.52 (↓ -3.11) | 56.69 (↓ -4.36) | 41.25 (↓ -1.45) |
| | | IDF | | 64.36 | 55.98 | 64.64 | 45.31 |
| | top 1 | | | 58.30 (↓ -6.06) | 49.73 (↓ -6.25) | 56.22 (↓ -8.42) | 37.72 (↓ -7.59) |
| | 5-12 | | | 62.85 (↓ -1.51) | 53.83 (↓ -2.15) | 61.26 (↓ -3.38) | 44.62 (↓ -0.69) |
| | 1-12 | | | 62.53 (↓ -1.83) | 53.38 (↓ -2.60) | 61.60 (↓ -3.04) | 45.42 (↑ 0.11) |
| WRD | | norm | cosine | 69.73 | 71.30 | 67.06 | 47.85 |
| | top 1 | | | 68.57 (↓ -1.16) | 70.23 (↓ -1.07) | 64.09 (↓ -2.97) | 44.35 (↓ -3.50) |
| | 5-12 | | | 69.54 (↓ -0.19) | 71.13 (↓ -0.17) | 65.67 (↓ -1.39) | 46.14 (↓ -1.71) |
| | 1-12 | | | 69.47 (↓ -0.26) | 71.09 (↓ -0.21) | 65.72 (↓ -1.34) | 46.26 (↓ -1.59) |
| | | IDF | | 69.89 | 72.23 | 67.85 | 50.67 |
| | top 1 | | | 68.65 (↓ -1.24) | 71.45 (↓ -0.78) | 65.16 (↓ -2.69) | 47.34 (↓ -3.33) |
| | 5-12 | | | 69.65 (↓ -0.24) | 72.16 (↓ -0.07) | 66.62 (↓ -1.23) | 49.08 (↓ -1.59) |
| | 1-12 | | | 69.56 (↓ -0.33) | 72.14 (↓ -0.09) | 66.69 (↓ -1.16) | 49.18 (↓ -1.49) |
| SynWMD | | SWF | cosine | 71.56 | 73.90 | 69.30 | 54.21 |
| | top 1 | | | 67.79 (↓ -3.77) | 68.76 (↓ -5.14) | 64.20 (↓ -5.10) | 44.35 (↓ -9.86) |
| | 5-12 | | | 71.00 (↓ -0.56) | 72.36 (↓ -1.54) | 67.03 (↓ -2.27) | 49.75 (↓ -4.46) |
| | 1-12 | | | 70.69 (↓ -0.87) | 72.05 (↓ -1.85) | 67.10 (↓ -2.20) | 50.02 (↓ -4.19) |
| | | | SWD | 70.86 | 74.67 | 68.52 | 57.09 |
| | top 1 | | | 68.30 (↓ -2.56) | 71.23 (↓ -3.44) | 65.54 (↓ -2.98) | 49.22 (↓ -7.87) |
| | 5-12 | | | 70.76 (↓ -0.10) | 73.52 (↓ -1.15) | 67.32 (↓ -1.20) | 54.02 (↓ -3.07) |
| | 1-12 | | | 70.55 (↓ -0.31) | 73.31 (↓ -1.36) | 67.34 (↓ -1.18) | 54.34 (↓ -2.75) |

Table 25: The scores on the STSB test set for the methods using the embeddings from DistilBERT and $\text{BERT}_{\text{seed0}}$. For WMD, WRD, and SynWMD, the scores of WSMD and the original method are compared for each layer selection. Score increases are highlighted in red and score decreases are highlighted in blue. Maximum scores for each embedding are underlined.

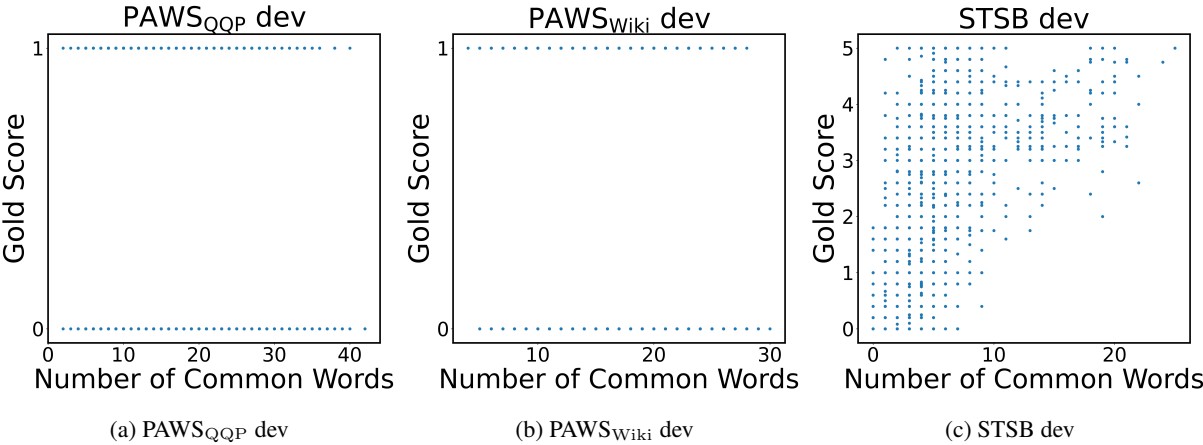

(a) $\text{PAWS}_{\text{QQP}}$ dev

(b) $\text{PAWS}_{\text{Wiki}}$ dev

(c) STSB dev

Figure 10: Scatter plots of the number of common words in sentence pairs for each dataset against the corresponding gold scores. (a) Spearman's $\rho \times 100 = -9.31$ for the $\text{PAWS}_{\text{QQP}}$ dev set. (b) Spearman's $\rho \times 100 = -0.85$ for the $\text{PAWS}_{\text{Wiki}}$ dev set. (c) Spearman's $\rho \times 100 = 59.37$ for STSB dev set.

|  | Test |
| --- | --- |
| SICK-R | 4927 |

Table 26: The number of sentence pairs in the SICK-R test set.

| methods | layer | weight | cost | SICK-R (Spearman's $\rho \times 100$) BERT0 | BERT12 |
| --- | --- | --- | --- | --- | --- |
| Avg. Pool. | | | | 44.10 | 62.19 |
| BERTScore | | uniform | | 50.29 | 58.55 |
| | | IDF | | 50.54 | 58.26 |
| DynaMax | | | | 48.38 | 61.99 |
| SIF | | | | 43.02 | 61.95 |
| uSIF | | | | 42.93 | 61.92 |
| Con. Neg. | | | | 44.34 | 62.30 |
| ROTS | | | | 44.46 | 62.26 |
| OPWD | | uniform | $L_2$ | 49.97 | 54.06 |
| | | | cosine | 48.49 | 61.72 |
| WMDo | | uniform | cosine | 36.93 | 60.70 |
| WMD | | uniform | $L_2$ | 49.97 | 54.06 |
| | top 1 | uniform | | 50.13 (↑ 0.16) | 53.39 (↓ -0.67) |
| | 5-12 | | | 50.14 (↑ 0.17) | 53.41 (↓ -0.65) |
| | 1-12 | | | 50.07 (↑ 0.10) | 53.04 (↓ -1.02) |
| | | IDF | | 49.39 | 53.55 |
| | top 1 | IDF | | 49.58 (↑ 0.19) | 52.99 (↓ -0.56) |
| | 5-12 | | | 49.53 (↑ 0.14) | 52.94 (↓ -0.61) |
| | 1-12 | | | 49.48 (↑ 0.09) | 52.57 (↓ -0.98) |
| WRD | | norm | cosine | 47.78 | 61.62 |
| | top 1 | norm | | 47.77 (↓ -0.01) | 61.51 (↓ -0.11) |
| | 5-12 | | | 47.75 (↓ -0.03) | 61.48 (↓ -0.14) |
| | 1-12 | | | 47.69 (↓ -0.09) | 61.34 (↓ -0.28) |
| | | IDF | | 48.13 | 61.29 |
| | top 1 | IDF | | 48.08 (↓ -0.05) | 61.20 (↓ -0.09) |
| | 5-12 | | | 48.05 (↓ -0.08) | 61.15 (↓ -0.14) |
| | 1-12 | | | 47.98 (↓ -0.15) | 61.02 (↓ -0.27) |
| SynWMD | | SWF | cosine | 50.52 | 63.35 |
| | top 1 | | cosine | 50.65 (↑ 0.13) | 62.49 (↓ -0.86) |
| | 5-12 | | | 50.47 (↓ -0.05) | 62.35 (↓ -1.00) |
| | 1-12 | | | 50.38 (↓ -0.14) | 61.97 (↓ -1.38) |
| | | | SWD | 49.59 | 63.54 |
| | top 1 | | SWD | 49.92 (↑ 0.33) | 62.96 (↓ -0.58) |
| | 5-12 | | | 49.70 (↑ 0.11) | 62.78 (↓ -0.76) |
| | 1-12 | | | 49.70 (↑ 0.11) | 62.52 (↓ -1.02) |

Table 27: The scores on the SICK-R test set for the methods using the BERT embeddings. For WMD, WRD, and SynWMD, the scores of WSMD and the original method are compared for each layer selection. Score increases are highlighted in red and score decreases are highlighted in blue. Maximum scores for each embedding are underlined.

|  | Train |
| --- | --- |
| PAWS$_{QQP}$ | 11988 (31.50%) |

Table 28: The number of sentence pairs in the PAWS$_{QQP}$ train set. The percentage of paraphrases is shown in parentheses.

| methods | layer | weight | cost | PAWS$_{QQP}$ train (AUC × 100) | | | | | |
|---|---|---|---|---|---|---|---|---|---|
| | | | | BERT0 | BERT12 | SimCSE0 | SimCSE12 | RoBERTa0 | RoBERTa12 |
| Avg. Pool. | | | | 66.14 | 75.57 | 66.84 | 73.92 | 61.58 | 73.73 |
| BERTScore | | uniform | | 50.41 | 58.79 | 50.61 | 52.59 | 45.65 | 57.97 |
| | | IDF | | 50.74 | 58.51 | 50.73 | 52.52 | 48.21 | 59.33 |
| DynaMax | | | | 67.61 | 79.25 | 68.12 | 73.42 | 63.56 | 71.49 |
| SIF | | | | 65.80 | 75.28 | 66.24 | 73.34 | 61.58 | 73.04 |
| uSIF | | | | 65.89 | 75.28 | 66.22 | 73.36 | 61.67 | 72.99 |
| Con. Neg. | | | | 66.03 | 75.64 | 66.48 | 74.04 | 61.60 | 73.69 |
| ROTS | | | | 60.46 | 76.59 | 61.13 | 74.55 | 54.91 | 74.01 |
| OPWD | | uniform | $L_2$ | 67.65 | 77.90 | 67.77 | 75.49 | 64.35 | 76.98 |
| | | | cosine | 68.92 | 79.73 | 68.80 | 75.91 | 66.21 | 76.46 |
| WMDo | | uniform | cosine | 49.84 | 61.95 | 49.85 | 53.28 | 59.53 | 68.86 |
| WMD | | | | 67.65 | 77.90 | 67.77 | 75.49 | 64.35 | 76.98 |
| | top 1* | uniform | | 79.42 (↑ 11.77) | 78.43 (↑ 0.53) | 74.81 (↑ 7.04) | 76.04 (↑ 0.55) | 73.61 (↑ 9.26) | 77.02 (↑ 0.04) |
| | 5-12 | | | 76.49 (↑ 8.84) | 78.13 (↑ 0.23) | 75.11 (↑ 7.34) | 75.75 (↑ 0.26) | 73.70 (↑ 9.35) | 76.98 (–) |
| | 1-12 | | $L_2$ | 76.49 (↑ 8.84) | 78.10 (↑ 0.20) | 76.13 (↑ 8.36) | 76.04 (↑ 0.55) | 73.06 (↑ 8.71) | 76.65 (↓ -0.33) |
| | | | | 67.41 | 77.54 | 67.53 | 75.28 | 64.67 | 76.12 |
| | top 1* | IDF | | 78.88 (↑ 11.47) | 78.16 (↑ 0.62) | 75.22 (↑ 7.69) | 75.86 (↑ 0.58) | 73.03 (↑ 8.36) | 76.26 (↑ 0.14) |
| | 5-12 | | | 75.88 (↑ 8.47) | 77.80 (↑ 0.26) | 74.66 (↑ 7.13) | 75.53 (↑ 0.25) | 73.05 (↑ 8.38) | 76.08 (↓ -0.04) |
| | 1-12 | | | 75.99 (↑ 8.58) | 77.79 (↑ 0.25) | 75.85 (↑ 8.32) | 75.84 (↑ 0.56) | 72.43 (↑ 7.76) | 75.70 (↓ -0.42) |

Table 29: The scores on the PAWS$_{QQP}$ train set for the methods using the embeddings from BERT, SimCSE, and RoBERTa. For WMD, the scores of WSMD and the original method are compared for each layer selection. Score increases are highlighted in red and score decreases are highlighted in blue. Maximum scores for each embedding are underlined. Note that the top layer was determined based on the first 1500 sentence pairs. For this reason, *top1* is marked with an asterisk.