# OpenReview forum: "Improving word mover's distance by leveraging self-attention matrix"
_EMNLP/2023/Conference — EMNLP 2023 Findings_

### Official Review · Reviewer_RMgW · 2023-07-30

**Typos Grammar Style And Presentation Improvements:** Line286
**Soundness:** 3

**Excitement:**

3: Ambivalent: It has merits (e.g., it reports state-of-the-art results, the idea is nice), but there are key weaknesses (e.g., it describes incremental work), and it can significantly benefit from another round of revision. However, I won't object to accepting it if my co-reviewers champion it.

**Paper Topic And Main Contributions:**

This paper tackled the problem of WMDs, which does not consider the order of words in a sentence.
Thus, the authors propose a method to incorporate GW and Self-attention matrix of pretrained BERT, which is called as SMD.
Furthermore, they combine WMD and SMD with a hyperparameter, denoted by WSMD.

However, I can hardly understand the results and the explanations.

--- After rebuttal ---

I updated my scores toward positive. Thanks for the clear explanation and I hope those explanations will be included in the final version.

**Questions For The Authors:**

- Will other variations of BERT also comply the idea of using Self-Attention matrix for measuring semantic similarity?
  - If yes, I think the metric contributes a lot to NLP field (but the authors should show the result)
  - Otherwise, the contribution is limited to be applicable in BERT model.

**Reasons To Accept:**

- The method is simple and straightforward

**Reasons To Reject:**

- In writing, the paper is hard to understand
  - In particular, Figure 4 and Figure 6 seem important to support the method, but it is not readable in this state.
  - I recommend the authors to move ablation parts (e.g., selecting BERT's head) to Appendix and focus on highlightening their method in the main part.

- No comparison with other general methods to measure semantic similarity
  - Although I can see the method shows better score than the traditional methods,
  - at least, BERTscore should be presented.

**Reproducibility:**

4: Could mostly reproduce the results, but there may be some variation because of sample variance or minor variations in their interpretation of the protocol or method.

**Reviewer Confidence:**

2: Willing to defend my evaluation, but it is fairly likely that I missed some details, didn't understand some central points, or can't be sure about the novelty of the work.

---

> ### Author Rebuttal · Authors · 2023-08-29
>
> Thank you for your comments and suggestions for improving the paper. Below are our responses:
>
> ---
>
> # Question about the paper readability
>
> > In writing, the paper is hard to understand. In particular, Figure 4 and Figure 6 seem important to support the method, but it is not readable in this state.
>
>
> A1. We appreciate your insightful suggestions. Indeed, we acknowledge that our paper contained sections that may have been difficult to understand, including Fig. 4 and Fig. 6. Therefore, we will review these sections and figures, and update the paper to improve clarity in the camera-ready version.
>
> ### Brief explanation of our proposed method (please skip this part if you're already familiar with it)
> First, while the Wasserstein distance-based WMD is used to measure sentence similarity, it treats sentences as sets. This means that it cannot distinguish between two semantically different sentences such as
>
> - The President greets the press in Chicago.
> - The press greets the President in Chicago.
>
> To address this problem, we focused on the Fused Gromov-Wasserstein distance, which combines the Wasserstein distance with the structural information provided by the Gromov-Wasserstein distance. We proposed the WSMD, which uses a WMD-like method for the Wasserstein distance term, and the Self-Attention Matrix (SAM), which encodes sentence structure information, for the Gromov-Wasserstein term. We believe that **our approach is novel for two reasons**:
>
> 1. The use of the Fused Gromov-Wasserstein distance for sentence similarity is novel. While WMD based on Wasserstein distance is well known, to our knowledge, no method has been proposed to measure sentence similarity using the Fused Gromov-Wasserstein distance.
> 2. Demonstrate high performance using SAM in the Gromov-Wasserstein term. Optimal transport distances allow for custom distribution and cost function definitions, providing significant research value. For example,  WRD, which improves on WMD, defines its distribution and cost function using word embedding length and cosine similarity, and is known to work effectively.
>
> ### Explanation of Fig. 4 and Fig. 6
> Regarding Fig. 4, BERT has 12 layers and 12 heads, resulting in 144 SAM options for WSMD. **Fig. 4 is a scatter plot showing the AUC for both PAWSQQP and PAWSWiki calculated using WSMD for these 144 SAMs.** Since each layer contains 12 heads, we also show the scores averaged over the WSMDs using these heads. Performance varies significantly between different heads because they capture different information, but when averaged across a layer, the variability is reduced. Therefore, considering its use in other datasets such as STSB, we decided to select a layer, compute the WSMD using SAMs within that layer, and evaluate performance based on the average of those values. I will discuss the layer in more detail in response to A2.
>
> Finally, **Fig. 6 provides a visual representation of the results in Tables 18, 19, and 20 of Appendix F.** For WMD, WRD, and SynWMD, we set their respective scores as **Baseline Score** and the scores with WSMD as **Score with WSMD**. If a method is above the diagonal line, it indicates improved performance with WSMD. Note that certain baselines, such as SIF, cannot use WSMD, so they are plotted on the diagonal line.
>
> ---
>
> # Recommendation to move ablation parts to the appendix
>
> > I recommend the authors to move ablation parts (e.g., selecting BERT's head) to Appendix and focus on highlightening their method in the main part.
>
> A2. We appreciate your insightful suggestions. We will rewrite our paper to make it more reader-friendly and may incorporate these changes in the camera-ready version. Thank you for your constructive feedback.
>
> To provide more context, we explain the reason why Section 7.5 on ablation parts is in the main text. While WMD is a fully unsupervised method, WSMD requires the selection of SAM and lambda. So, we set lambda to 0.5, and if a dev set is available, we use the SAMs from the best performing layer on the dev set for WSMD.
>
> In cases without a dev set, we propose a fully unsupervised method to select multiple layers and use their SAMs for WSMD. The most naive layer selection in the absence of a dev set would be to use all layers. However, in order to select layers with more structural information, we defined BV_A and also presented results when selecting layers 5-12, which have high BV_A values.
>
> Given this context, we felt that the method of selecting the BERT head was important enough to be included in the main text, although it may be worth moving Section 7.5 to the Appendix to give more focus to our method.
>
> ---
>
> # Question about no comparison with other general methods
>
> > No comparison with other general methods to measure semantic similarity. Although I can see the method shows better score than the traditional methods, at least, BERTScore should be presented.
>
> A3.  Thank you for highlighting this issue. Your point about preparing an appropriate baseline to argue for the empirical performance of our proposed method convincingly is well taken. We also appreciate the suggestion of BERTScore, a relevant prior work that measures sentence similarity based on word-to-word correspondences.
>
> We then conducted comparative experiments using BERTScore as a baseline.  We find that **the performance of BERTScore was not high compared to WSMD, especially on PAWS**. This is due to the fact that BERTScore, like WMD and SIF, treats sentences as sets. This method struggles to identify paraphrases with high overlap. While BERTScore is widely used as an automatic evaluation metric for generated sentences, our results suggest that there is room for improvement in BERTScore.
>
> ### Experimental details (If you are interested in the details of the experiments, please take a look.)
> The DistilBERT results on PAWSQQP, PAWSWiki, and STSB were as follows:
>
> ### PAWSQQP
> | method | weight | BERT0 | BERT12 | SimCSE0 | SimCSE12 | RoBERTa0 | RoBERTa12 |
> |---|---|---|---|---|---|---|---|
> | BERTScore | uniform | 45.52 | 54.78 | 45.41 | 51.74 | 41.91 | 51.61 |
> | BERTScore | idf | 45.40 | 55.29 | 45.25 | 51.98 | 42.71 | 53.01 |
>
> ### PAWSWiki
> | method | weight | BERT0 | BERT12 | SimCSE0 | SimCSE12 | RoBERTa0 | RoBERTa12 |
> |---|---|---|---|---|---|---|---|
> | BERTScore | uniform | 52.16 | 62.45 | 52.15 | 56.09 | 53.26 | 60.34 |
> | BERTScore | idf | 52.29 | 62.53 | 52.27 | 56.10 | 51.96 | 59.29 |
>
> ### STSB
> | method | weight | BERT0 | BERT12 | SimCSE0 | SimCSE12 | RoBERTa0 | RoBERTa12 |
> |---|---|---|---|---|---|---|---|
> | BERTScore | uniform | 63.27 | 65.13 | 62.90 | 70.59 | 63.38 | 58.71 |
> | BERTScore | idf | 63.99 | 65.96 | 63.72 | 70.96 | 64.09 | 59.82 |
>
> Using 1-12 layers, we then show the WSMD results of uniform weight WMD from Tables 18 and 19 on PAWS, and those of SWD cost SynWMD from Table 20 on STSB for unsupervised comparison with BERTScore:
>
> ### PAWSQQP
> | method | layer | weight | cost | BERT0 | BERT12 | SimCSE0 | SimCSE12 | RoBERTa0 | RoBERTa12 |
> |---|---|---|---|---|---|---|---|---|---|
> |WMD|1-12|uniform|$L_2$|59.41|66.71|57.54|66.13|58.33 |66.76|
>
> ### PAWSWiki
> | method | layer | weight | cost | BERT0 | BERT12 | SimCSE0 | SimCSE12 | RoBERTa0 | RoBERTa12 |
> |---|---|---|---|---|---|---|---|---|---|
> |WMD|1-12|uniform|$L_2$|61.95 | 71.72 | 59.52| 66.50 | 62.71| 69.99 |
>
> ### STSB
> | method | layer | weight | cost | BERT0 | BERT12 | SimCSE0 | SimCSE12 | RoBERTa0 | RoBERTa12 |
> |---|---|---|---|---|---|---|---|---|---|
> | SynWMD | 1-12 | SWF | SWD | 71.73 | 71.97| 66.96 | 78.05 | 71.79 | 68.98|
>
> These results show that the performance of BERTScore was not high compared to WSMD, especially on PAWS. These experimental results will be included in the camera-ready version. We appreciate your constructive comments.
>
> ---
>
> # Quesiton about the applicablity of the method for other BERT-based models
>
> > Will other variations of BERT also comply the idea of using Self-Attention matrix for measuring semantic similarity?
>
> A4. Thank you for highlighting this issue. We believe the answer is yes. We have done similar experiments with DistilBERT, which has a different number of layers compared to BERT and RoBERTa, and **observed performance improvements with WSMD on PAWS**.
> This suggests that WSMD may be applicable to a wider range of models than just BERT and RoBERTa.
>
> ### Experimental details (If you are interested in the details of the experiments, please take a look.)
> DistilBERT is a light model with 6 layers and 12 heads, distilled from BERT. The BV_A values were as follows:
> |Layer|1|2|3|4|5|6|
> |---|---|---|---|---|---|---|
> |DistilBERT-$BV_A$|0.293|0.267|0.461|0.499|0.475|0.471|
>
> Although BERT and RoBERTa have 12 layers and DistilBERT has only 6 layers, the BV_A per layer, as seen in Fig. 5, starts low for the initial layers and increases thereafter. Therefore, we chose the layers top1, 3-6, 1-6 for WSMD in the DistilBERT experiments. The top1 layer was chosen based on the AUC from averaging the WSMD over a single layer starting from the 3rd layer using the PAWSQQP dev set, and the layers for the 0th and 6th embeddings were chosen as follows:
> ||DistilBERT|
> |---|---|
> |0th Embs. |4|
> |6th Embs. |4|
>
> Then, the results on PAWSQQP, PAWSWiki, and STSB were as follows:
> ### PAWSQQP
> | method | layer | weight | cost | DistilBERT0 | DistilBERT6 |
> |---|---|---|---|---|---|
> | Avg. Pool. |  |  |  | 52.02 | 63.22 |
> | SIF |  | sif |  | 52.26 | 63.03 |
> | uSIF |  | usif |  | 52.29 | 63.06 |
> | Con. Neg. |  | sif |  | 52.15 | 63.12 |
> | BERTScore |  | uniform |  | 45.58 | 54.74 |
> | BERTScore |  | IDF |  | 45.46 | 55.30 |
> | ROTS |  | sif |  | 51.71 | 64.95 |
> | OPWD |  | uniform | $L_2$ | 53.03 | 67.33 |
> | OPWD |  | uniform | cosine | 54.73 | **70.93** |
> | WMDo |  | uniform | cosine | 44.18 | 55.42 |
> | WMD |  | uniform | $L_2$ | 53.03 | 67.33 |
> | WMD | top1 | uniform | $L_2$ | 61.53 (+8.50) | 67.77 (+0.44) |
> | WMD | 3-6 | uniform | $L_2$ | 58.69 (+5.66) | 67.64 (+0.31) |
> | WMD | 1-6 | uniform | $L_2$ | 59.01 (+5.98) | 67.38 (+0.05) |
> | WMD |  | IDF | $L_2$ | 52.44 | 66.79 |
> | WMD | top1 | IDF | $L_2$ | 60.38 (+7.94) | 67.32 (+0.53) |
> | WMD | 3-6 | IDF | $L_2$ | 57.90 (+5.46) | 67.13 (+0.34) |
> | WMD | 1-6 | IDF | $L_2$ | 58.07 (+5.63) | 66.87 (+0.08) |
> | WRD |  | norm | cosine | 53.25 | 70.39 |
> | WRD | top1 | norm | cosine | 65.44 (+12.19) | 70.66 (+0.27) |
> | WRD | 3-6 | norm | cosine | 62.57 (+9.32) | 70.58 (+0.19) |
> | WRD | 1-6 | norm | cosine | 62.95 (+9.70) | 70.50 (+0.11) |
> | WRD |  | IDF | cosine | 54.46 | 70.59 |
> | WRD | top1 | IDF | cosine | **66.81** (+12.35) | **70.93** (+0.34) |
> | WRD | 3-6 | IDF | cosine | 65.07 (+10.61) | 70.82 (+0.23) |
> | WRD | 1-6 | IDF | cosine | 64.97 (+10.51) | 70.71 (+0.12) |
> | SynWMD |  | SWF | cosine | 47.43 | 60.98 |
> | SynWMD | top1 | SWF | cosine | 58.71 (+11.28) | 61.07 (+0.09) |
> | SynWMD | 3-6 | SWF | cosine | 56.40 (+8.97) | 60.72 (-0.26) |
> | SynWMD | 1-6 | SWF | cosine | 55.76 (+8.33) | 60.17 (-0.81) |
> | SynWMD |  | SWF | SWD | 58.43 | 64.47 |
> | SynWMD | top1 | SWF | SWD | 61.51 (+3.08) | 63.99 (-0.48) |
> | SynWMD | 3-6 | SWF | SWD | 60.00 (+1.57) | 63.93 (-0.54) |
> | SynWMD | 1-6 | SWF | SWD | 60.06 (+1.63) | 63.50 (-0.97) |
>
> ### PAWSWiki
> | method | layer | weight | cost | DistilBERT0 | DistilBERT6 |
> |---|---|---|---|---|---|
> | Avg. Pool. |  |  |  | 53.36 | 60.17 |
> | SIF |  | sif |  | 53.25 | 60.20 |
> | uSIF |  | usif |  | 53.18 | 60.33 |
> | Con. Neg. |  | sif |  | 53.33 | 60.40 |
> | BERTScore |  | uniform |  | 52.06 | 59.78 |
> | BERTScore |  | IDF |  | 52.25 | 59.78 |
> | ROTS |  | sif |  | 56.73 | 62.68 |
> | OPWD |  | uniform | $L_2$ | 57.43 | 71.99 |
> | OPWD |  | uniform | cosine | 54.86 | 69.95 |
> | WMDo |  | uniform | cosine | 49.69 | 58.62 |
> | WMD |  | uniform | $L_2$ | 57.43 | 71.99 |
> | WMD | top1 | uniform | $L_2$ | **61.45** (+4.02) | 72.56 (+0.57) |
> | WMD | 3-6 | uniform | $L_2$ | 59.81 (+2.38) | 72.07 (+0.08) |
> | WMD | 1-6 | uniform | $L_2$ | 60.72 (+3.29) | 71.69 (-0.30) |
> | WMD |  | IDF | $L_2$ | 57.02 | 72.23 |
> | WMD | top1 | IDF | $L_2$ | 60.79 (+3.77) | **72.81** (+0.58) |
> | WMD | 3-6 | IDF | $L_2$ | 59.29 (+2.27) | 72.30 (+0.07) |
> | WMD | 1-6 | IDF | $L_2$ | 60.11 (+3.09) | 71.78 (-0.45) |
> | WRD |  | norm | cosine | 55.11 | 70.48 |
> | WRD | top1 | norm | cosine | 58.68 (+3.57) | 70.89 (+0.41) |
> | WRD | 3-6 | norm | cosine | 57.51 (+2.40) | 70.63 (+0.15) |
> | WRD | 1-6 | norm | cosine | 58.43 (+3.32) | 70.65 (+0.17) |
> | WRD |  | IDF | cosine | 54.38 | 69.77 |
> | WRD | top1 | IDF | cosine | 58.39 (+4.01) | 70.21 (+0.44) |
> | WRD | 3-6 | IDF | cosine | 57.17 (+2.79) | 69.91 (+0.14) |
> | WRD | 1-6 | IDF | cosine | 58.17 (+3.79) | 69.90 (+0.13) |
> | SynWMD |  | SWF | cosine | 50.43 | 60.70 |
> | SynWMD | top1 | SWF | cosine | 53.94 (+3.51) | 60.78 (+0.08) |
> | SynWMD | 3-6 | SWF | cosine | 52.47 (+2.04) | 60.42 (-0.28) |
> | SynWMD | 1-6 | SWF | cosine | 53.08 (+2.65) | 59.96 (-0.74) |
> | SynWMD |  | SWF | SWD | 52.94 | 60.52 |
> | SynWMD | top1 | SWF | SWD | 54.90 (+1.96) | 60.36 (-0.16) |
> | SynWMD | 3-6 | SWF | SWD | 53.95 (+1.01) | 60.11 (-0.41) |
> | SynWMD | 1-6 | SWF | SWD | 54.13 (+1.19) | 59.67 (-0.85) |
>
> ### STSB
> | method | layer | weight | cost | DistilBERT0 | DistilBERT6 |
> |---|---|---|---|---|---|
> | Avg. Pool. |  |  |  | 66.33 | 71.83 |
> | SIF |  | sif |  | 65.35 | 71.76 |
> | uSIF |  | usif |  | 65.25 | 71.64 |
> | Con. Neg. |  | sif |  | 65.81 | 71.86 |
> | BERTScore |  | uniform |  | 63.30 | 63.01 |
> | BERTScore |  | IDF |  | 64.16 | 63.92 |
> | ROTS |  | sif |  | 66.61 | 71.98 |
> | OPWD |  | uniform | $L_2$ | 60.59 | 52.63 |
> | OPWD |  | uniform | cosine | 67.29 | 70.93 |
> | WMDo |  | uniform | cosine | 66.59 | 70.13 |
> | WMD |  | uniform | $L_2$ | 60.59 | 52.63 |
> | WMD | top1 | uniform | $L_2$ | 58.88 (-1.71) | 50.20 (-2.43) |
> | WMD | 3-6 | uniform | $L_2$ | 59.24 (-1.35) | 50.66 (-1.97) |
> | WMD | 1-6 | uniform | $L_2$ | 58.07 (-2.52) | 49.52 (-3.11) |
> | WMD |  | IDF | $L_2$ | 64.36 | 55.98 |
> | WMD | top1 | IDF | $L_2$ | 63.11 (-1.25) | 53.90 (-2.08) |
> | WMD | 3-6 | IDF | $L_2$ | 63.39 (-0.97) | 54.25 (-1.73) |
> | WMD | 1-6 | IDF | $L_2$ | 62.53 (-1.83) | 53.38 (-2.60) |
> | WRD |  | norm | cosine | 69.73 | 71.30 |
> | WRD | top1 | norm | cosine | 69.57 (-0.16) | 71.15 (-0.15) |
> | WRD | 3-6 | norm | cosine | 69.63 (-0.10) | 71.19 (-0.11) |
> | WRD | 1-6 | norm | cosine | 69.47 (-0.26) | 71.09 (-0.21) |
> | WRD |  | IDF | cosine | 69.89 | 72.23 |
> | WRD | top1 | IDF | cosine | 69.68 (-0.21) | 72.16 (-0.07) |
> | WRD | 3-6 | IDF | cosine | 69.74 (-0.15) | 72.20 (-0.03) |
> | WRD | 1-6 | IDF | cosine | 69.56 (-0.33) | 72.14 (-0.09) |
> | SynWMD |  | SWF | cosine | **71.56** | 73.90 |
> | SynWMD | top1 | SWF | cosine | 71.17 (-0.39) | 72.20 (-1.70) |
> | SynWMD | 3-6 | SWF | cosine | 71.39 (-0.17) | 72.69 (-1.21) |
> | SynWMD | 1-6 | SWF | cosine | 70.69 (-0.87) | 72.05 (-1.85) |
> | SynWMD |  | SWF | SWD | 70.86 | **74.67** |
> | SynWMD | top1 | SWF | SWD | 70.89 (+0.03) | 73.26 (-1.41) |
> | SynWMD | 3-6 | SWF | SWD | 71.02 (+0.16) | 73.65 (-1.02) |
> | SynWMD | 1-6 | SWF | SWD | 70.55 (-0.31) | 73.31 (-1.36) |
>
> These experimental results will be included in the camera-ready version. We appreciate your constructive comments.
>
> ---
>
> # Question about typos
>
> > Line286: m->j
>
> A5. We apologize for any confusion caused by unclear explanations. We would like to clarify that the point raised was **not a typographical error**. Such feedback is invaluable in helping us to produce a more accessible manuscript for readers.
>
> We will provide a detailed explanation of the point raised. Regarding the comment about whether $m$ is a typo for $j$ on line 286, the notation $P_{ij}$ represents the transport amount from the $i$-th word of sentence $s$ to the $j$-th word of sentence $s'$, and $p_i=(P_{i1},\cdots,P_{im})$ represents the array of transport amounts from the $i$-th word of $s$ to the $j$-th word of $s’$, where $j=1,\cdots,m$. As shown in Fig. 3, the equation $\sum_{j=1}^m P_{ij} = u_i$ holds. We acknowledge the complexity of the notation and will try to provide a clearer explanation.

---

### Official Review · Reviewer_ycfD · 2023-08-04

**Soundness:** 3

**Excitement:**

4: Strong: This paper deepens the understanding of some phenomenon or lowers the barriers to an existing research direction.

**Paper Topic And Main Contributions:**

The paper mainly focuses on improving the Word Mover's Distance (WMD) method for short text representation. A new Word and Sentence Mover's Distance (WSMD) is proposed and validated to optimize both word vector distance and sentence structure distance, in order to more accurately characterize sentence semantics.
The main contributions are:

The Word Mover's Distance (WMD) method cannot consider word order information when calculating sentence similarity, which may result in high similarity scores for sentences with different meanings. The proposed Word and Sentence Mover's Distance (WSMD) method simultaneously utilizes both word vector distance and sentence structure distance, allowing sentence structure information to participate in the calculation of similarity.

The authors introduced the self-attention matrix (SAM) output by the BERT model to represent sentence structure and designed the SAM-based Sentence Structure Distance (SMD).

They also proposed the WSMD distance metric, which simultaneously optimizes both word vector distance and sentence structure distance. The WSMD effectively combines word vector information and sentence structure information to compute sentence similarity.

The effectiveness of WSMD was validated on the PAWS dataset for sentence similarity judgment tasks, and the results outperformed multiple baselines. This provides a new approach for sentence representation, characterizing sentence semantics through matching word vectors and syntactic structures.

**Reasons To Accept:**

A new idea was proposed to use the self-attention mechanism of BERT to represent the syntactic structure information of sentences, which is an aspect that has been less considered in previous work on short text representation.

The WSMD method was developed by adding sentence structure matching to the existing word vector similarity, optimizing both word vector distance and sentence structure distance, thus improving the quality of sentence representation.

The SMD based on sentence self-attention matrix was designed, providing an innovative way to measure structural similarity.

The effectiveness of the proposed method was thoroughly verified on the PAWS dataset for sentence similarity tasks, and the results show that WSMD outperforms the baseline methods, demonstrating its effectiveness.

**Reasons To Reject:**

The improvement in performance on the STS Benchmark dataset for semantic text similarity was very limited.

Different pre-trained language models use different tokenization methods, which limits the transferability of the proposed method and its general applicability.

The size and scope of the two datasets used in the paper were relatively small. If the proposed method could be validated on more diverse and large-scale datasets, the results would be more convincing.

**Reproducibility:**

4: Could mostly reproduce the results, but there may be some variation because of sample variance or minor variations in their interpretation of the protocol or method.

**Reviewer Confidence:**

4: Quite sure. I tried to check the important points carefully. It's unlikely, though conceivable, that I missed something that should affect my ratings.

---

> ### Author Rebuttal · Authors · 2023-08-29
>
> Thank you for your comments and suggestions for improving the paper. Below are our responses:
>
> ---
>
> # Question about the performance on STSB
>
> > The improvement in performance on the STS Benchmark dataset for semantic text similarity was very limited.
>
> A1. Thank you for highlighting this issue. Indeed, the performance improvement on the STS semantic text similarity benchmark dataset was quite limited. In the future, we are considering dynamically calculating the lambda for each sentence pair to ensure that the GW term adjusts as needed.
>
> To provide more context, we explain the reason behind this issue. As can be seen in Fig. 9 in Appendix G, there is a trend in STSB where **the gold score increases as the number of common words increases**. As a result, methods that treat sentences as sets of words can achieve good performance, and in WSMD, the GW term can potentially act as noise.
>
> ---
>
> # Question about the method limitation of the transferability and the general applicability
>
> > Different pre-trained language models use different tokenization methods, which limits the transferability of the proposed method and its general applicability.
>
> A2. Thank you for highlighting this issue. Certainly, different pre-trained language models use different tokenization methods, which challenges the transferability of our proposed method and its general applicability. We believe that **WSMD can address this issue by mapping tokens derived from each tokenizer**.
>
> For example, both SynWMD and ROTS use different tokenizers than BERT to derive parsing trees from sentences. They address this problem in their implementations by using the mapping strategy.
>
> ---
>
> # Question about the evaluation dataset size
>
> > The size and scope of the two datasets used in the paper were relatively small. If the proposed method could be validated on more diverse and large-scale datasets, the results would be more convincing.
>
> A3. Thank you for highlighting this issue. Indeed, the size and scope of the two datasets used in our paper were relatively small.
> **We intend to repeat the experiments** using the full PAWS dataset and other STS datasets such as STS12-16 and SICKR,  and include the results in the camera-ready version as long as time permits. Thank you again for your constructive feedback.
>
> To provide more context,  I explain the reason. The original dataset sizes for PAWSQQP, PAWSWiki, and STSB are as follows:
>
> |Dataset|Dev|Test|
> |---|---|---|
> |PAWSQQP|11988|677|
> |PAWSWiki|8000|8000|
> |STSB|1500|1379|
>
> Since the data sizes for the PAWSQQP and PAWSWiki test sets are significantly different in our study, we adjusted them for experimental efficiency. We adjusted the sizes of the PAWSQQP dev and PAWSWiki dev and test sets to the STSB dev size of 1500 because STSB is one of the most widely used datasets and we felt that the size of 1500 sentence pairs was not too small, although the PAWSQQP test set is relatively small.

---

### Official Review · Reviewer_YoLW · 2023-08-11

**Soundness:** 4

**Excitement:**

3: Ambivalent: It has merits (e.g., it reports state-of-the-art results, the idea is nice), but there are key weaknesses (e.g., it describes incremental work), and it can significantly benefit from another round of revision. However, I won't object to accepting it if my co-reviewers champion it.

**Paper Topic And Main Contributions:**

In this paper, the authors proposed a small modification for World Mover's Distance (WMD) to improve paraphrases identification, They found that the World Mover's Distance (WMD) does not consider word order, which poses difficulties in distinguishing between sentences that have a substantial overlap of similar words but divergent semantic meanings. Therefore, they introduced an enhancement to the WMD by integrating sentence structures as represented by BERT's self-attention matrix (SAM). The suggested approach is grounded on the Fused Gromov-Wasserstein distance, which concurrently factors in the word embedding similarity and the SAM when determining the optimal transport between two sentences. Experimental results reveal that the proposed approach improves upon the WMD and its variants on BERT, particularly in paraphrase identification, while maintaining near-equivalent performance in semantic textual similarity tasks.

**Reasons To Accept:**

- The authors proposed a small modification for WMD, and effective for paraphrase identification.
- The authors conducted a comprehensive empirical study on the proposed method.
- The paper is well-written and easy to understand.

**Reasons To Reject:**

- One limitation to note is that the authors exclusively tested their method using BERT and RoBERTa. An exploration into its generalizability across other leading language models and the ablation study of hyperparameters are not fully addressed.
- The evaluation dataset size is quite small.


**Reproducibility:**

4: Could mostly reproduce the results, but there may be some variation because of sample variance or minor variations in their interpretation of the protocol or method.

**Reviewer Confidence:**

3: Pretty sure, but there's a chance I missed something. Although I have a good feel for this area in general, I did not carefully check the paper's details, e.g., the math, experimental design, or novelty.

---

> ### Author Rebuttal · Authors · 2023-08-29
>
> Thank you for your comments and suggestions for improving the paper. Below are our responses:
>
> ---
>
> # Questions about the generalizability across other models and the ablation study
> > One limitation to note is that the authors exclusively tested their method using BERT and RoBERTa. An exploration into its generalizability across other leading language models and the ablation study of hyperparameters are not fully addressed.
>
> A1. Thank you for your valuable feedback. We acknowledge the concerns you've raised regarding the evaluation of our proposed method using only BERT and RoBERTa. We understand the need to evaluate its generalisability to other major language models and to perform a hyperparameter ablation study.
>
> At this point, we cannot confirm whether the performance of all models with self-attention improves when WSMD is used. We then conducted experiments with DistilBERT, a model different from BERT and RoBERTa, and **observed performance improvements on PAWS for DistilBERT**, similar to BERT and RoBERTa. This suggests that WSMD may be applicable to a wider range of models than just BERT and RoBERTa.
>
> ### Experimental details (If you are interested in the details of the experiments, please take a look.)
> DistilBERT is a light model with 6 layers and 12 heads, distilled from BERT. The BV_A values were as follows:
> |Layer|1|2|3|4|5|6|
> |---|---|---|---|---|---|---|
> |DistilBERT-$BV_A$|0.293|0.267|0.461|0.499|0.475|0.471|
>
> Although BERT and RoBERTa have 12 layers and DistilBERT has only 6 layers, the BV_A per layer, as seen in Fig. 5, starts low for the initial layers and increases thereafter. Therefore, we chose the layers top1, 3-6, 1-6 for WSMD in the DistilBERT experiments. The top1 layer was chosen based on the AUC from averaging the WSMD over a single layer starting from the 3rd layer using the PAWSQQP dev set, and the layers for the 0th and 6th embeddings were chosen as follows:
> ||DistilBERT|
> |---|---|
> |0th Embs. |4|
> |6th Embs. |4|
>
> Then, the results on PAWSQQP, PAWSWiki, and STSB were as follows:
> ### PAWSQQP
> | method | layer | weight | cost | DistilBERT0 | DistilBERT6 |
> |---|---|---|---|---|---|
> | Avg. Pool. |  |  |  | 52.02 | 63.22 |
> | SIF |  | sif |  | 52.26 | 63.03 |
> | uSIF |  | usif |  | 52.29 | 63.06 |
> | Con. Neg. |  | sif |  | 52.15 | 63.12 |
> | BERTScore |  | uniform |  | 45.58 | 54.74 |
> | BERTScore |  | IDF |  | 45.46 | 55.30 |
> | ROTS |  | sif |  | 51.71 | 64.95 |
> | OPWD |  | uniform | $L_2$ | 53.03 | 67.33 |
> | OPWD |  | uniform | cosine | 54.73 | **70.93** |
> | WMDo |  | uniform | cosine | 44.18 | 55.42 |
> | WMD |  | uniform | $L_2$ | 53.03 | 67.33 |
> | WMD | top1 | uniform | $L_2$ | 61.53 (+8.50) | 67.77 (+0.44) |
> | WMD | 3-6 | uniform | $L_2$ | 58.69 (+5.66) | 67.64 (+0.31) |
> | WMD | 1-6 | uniform | $L_2$ | 59.01 (+5.98) | 67.38 (+0.05) |
> | WMD |  | IDF | $L_2$ | 52.44 | 66.79 |
> | WMD | top1 | IDF | $L_2$ | 60.38 (+7.94) | 67.32 (+0.53) |
> | WMD | 3-6 | IDF | $L_2$ | 57.90 (+5.46) | 67.13 (+0.34) |
> | WMD | 1-6 | IDF | $L_2$ | 58.07 (+5.63) | 66.87 (+0.08) |
> | WRD |  | norm | cosine | 53.25 | 70.39 |
> | WRD | top1 | norm | cosine | 65.44 (+12.19) | 70.66 (+0.27) |
> | WRD | 3-6 | norm | cosine | 62.57 (+9.32) | 70.58 (+0.19) |
> | WRD | 1-6 | norm | cosine | 62.95 (+9.70) | 70.50 (+0.11) |
> | WRD |  | IDF | cosine | 54.46 | 70.59 |
> | WRD | top1 | IDF | cosine | **66.81** (+12.35) | **70.93** (+0.34) |
> | WRD | 3-6 | IDF | cosine | 65.07 (+10.61) | 70.82 (+0.23) |
> | WRD | 1-6 | IDF | cosine | 64.97 (+10.51) | 70.71 (+0.12) |
> | SynWMD |  | SWF | cosine | 47.43 | 60.98 |
> | SynWMD | top1 | SWF | cosine | 58.71 (+11.28) | 61.07 (+0.09) |
> | SynWMD | 3-6 | SWF | cosine | 56.40 (+8.97) | 60.72 (-0.26) |
> | SynWMD | 1-6 | SWF | cosine | 55.76 (+8.33) | 60.17 (-0.81) |
> | SynWMD |  | SWF | SWD | 58.43 | 64.47 |
> | SynWMD | top1 | SWF | SWD | 61.51 (+3.08) | 63.99 (-0.48) |
> | SynWMD | 3-6 | SWF | SWD | 60.00 (+1.57) | 63.93 (-0.54) |
> | SynWMD | 1-6 | SWF | SWD | 60.06 (+1.63) | 63.50 (-0.97) |
>
> ### PAWSWiki
> | method | layer | weight | cost | DistilBERT0 | DistilBERT6 |
> |---|---|---|---|---|---|
> | Avg. Pool. |  |  |  | 53.36 | 60.17 |
> | SIF |  | sif |  | 53.25 | 60.20 |
> | uSIF |  | usif |  | 53.18 | 60.33 |
> | Con. Neg. |  | sif |  | 53.33 | 60.40 |
> | BERTScore |  | uniform |  | 52.06 | 59.78 |
> | BERTScore |  | IDF |  | 52.25 | 59.78 |
> | ROTS |  | sif |  | 56.73 | 62.68 |
> | OPWD |  | uniform | $L_2$ | 57.43 | 71.99 |
> | OPWD |  | uniform | cosine | 54.86 | 69.95 |
> | WMDo |  | uniform | cosine | 49.69 | 58.62 |
> | WMD |  | uniform | $L_2$ | 57.43 | 71.99 |
> | WMD | top1 | uniform | $L_2$ | **61.45** (+4.02) | 72.56 (+0.57) |
> | WMD | 3-6 | uniform | $L_2$ | 59.81 (+2.38) | 72.07 (+0.08) |
> | WMD | 1-6 | uniform | $L_2$ | 60.72 (+3.29) | 71.69 (-0.30) |
> | WMD |  | IDF | $L_2$ | 57.02 | 72.23 |
> | WMD | top1 | IDF | $L_2$ | 60.79 (+3.77) | **72.81** (+0.58) |
> | WMD | 3-6 | IDF | $L_2$ | 59.29 (+2.27) | 72.30 (+0.07) |
> | WMD | 1-6 | IDF | $L_2$ | 60.11 (+3.09) | 71.78 (-0.45) |
> | WRD |  | norm | cosine | 55.11 | 70.48 |
> | WRD | top1 | norm | cosine | 58.68 (+3.57) | 70.89 (+0.41) |
> | WRD | 3-6 | norm | cosine | 57.51 (+2.40) | 70.63 (+0.15) |
> | WRD | 1-6 | norm | cosine | 58.43 (+3.32) | 70.65 (+0.17) |
> | WRD |  | IDF | cosine | 54.38 | 69.77 |
> | WRD | top1 | IDF | cosine | 58.39 (+4.01) | 70.21 (+0.44) |
> | WRD | 3-6 | IDF | cosine | 57.17 (+2.79) | 69.91 (+0.14) |
> | WRD | 1-6 | IDF | cosine | 58.17 (+3.79) | 69.90 (+0.13) |
> | SynWMD |  | SWF | cosine | 50.43 | 60.70 |
> | SynWMD | top1 | SWF | cosine | 53.94 (+3.51) | 60.78 (+0.08) |
> | SynWMD | 3-6 | SWF | cosine | 52.47 (+2.04) | 60.42 (-0.28) |
> | SynWMD | 1-6 | SWF | cosine | 53.08 (+2.65) | 59.96 (-0.74) |
> | SynWMD |  | SWF | SWD | 52.94 | 60.52 |
> | SynWMD | top1 | SWF | SWD | 54.90 (+1.96) | 60.36 (-0.16) |
> | SynWMD | 3-6 | SWF | SWD | 53.95 (+1.01) | 60.11 (-0.41) |
> | SynWMD | 1-6 | SWF | SWD | 54.13 (+1.19) | 59.67 (-0.85) |
>
> ### STSB
> | method | layer | weight | cost | DistilBERT0 | DistilBERT6 |
> |---|---|---|---|---|---|
> | Avg. Pool. |  |  |  | 66.33 | 71.83 |
> | SIF |  | sif |  | 65.35 | 71.76 |
> | uSIF |  | usif |  | 65.25 | 71.64 |
> | Con. Neg. |  | sif |  | 65.81 | 71.86 |
> | BERTScore |  | uniform |  | 63.30 | 63.01 |
> | BERTScore |  | IDF |  | 64.16 | 63.92 |
> | ROTS |  | sif |  | 66.61 | 71.98 |
> | OPWD |  | uniform | $L_2$ | 60.59 | 52.63 |
> | OPWD |  | uniform | cosine | 67.29 | 70.93 |
> | WMDo |  | uniform | cosine | 66.59 | 70.13 |
> | WMD |  | uniform | $L_2$ | 60.59 | 52.63 |
> | WMD | top1 | uniform | $L_2$ | 58.88 (-1.71) | 50.20 (-2.43) |
> | WMD | 3-6 | uniform | $L_2$ | 59.24 (-1.35) | 50.66 (-1.97) |
> | WMD | 1-6 | uniform | $L_2$ | 58.07 (-2.52) | 49.52 (-3.11) |
> | WMD |  | IDF | $L_2$ | 64.36 | 55.98 |
> | WMD | top1 | IDF | $L_2$ | 63.11 (-1.25) | 53.90 (-2.08) |
> | WMD | 3-6 | IDF | $L_2$ | 63.39 (-0.97) | 54.25 (-1.73) |
> | WMD | 1-6 | IDF | $L_2$ | 62.53 (-1.83) | 53.38 (-2.60) |
> | WRD |  | norm | cosine | 69.73 | 71.30 |
> | WRD | top1 | norm | cosine | 69.57 (-0.16) | 71.15 (-0.15) |
> | WRD | 3-6 | norm | cosine | 69.63 (-0.10) | 71.19 (-0.11) |
> | WRD | 1-6 | norm | cosine | 69.47 (-0.26) | 71.09 (-0.21) |
> | WRD |  | IDF | cosine | 69.89 | 72.23 |
> | WRD | top1 | IDF | cosine | 69.68 (-0.21) | 72.16 (-0.07) |
> | WRD | 3-6 | IDF | cosine | 69.74 (-0.15) | 72.20 (-0.03) |
> | WRD | 1-6 | IDF | cosine | 69.56 (-0.33) | 72.14 (-0.09) |
> | SynWMD |  | SWF | cosine | **71.56** | 73.90 |
> | SynWMD | top1 | SWF | cosine | 71.17 (-0.39) | 72.20 (-1.70) |
> | SynWMD | 3-6 | SWF | cosine | 71.39 (-0.17) | 72.69 (-1.21) |
> | SynWMD | 1-6 | SWF | cosine | 70.69 (-0.87) | 72.05 (-1.85) |
> | SynWMD |  | SWF | SWD | 70.86 | **74.67** |
> | SynWMD | top1 | SWF | SWD | 70.89 (+0.03) | 73.26 (-1.41) |
> | SynWMD | 3-6 | SWF | SWD | 71.02 (+0.16) | 73.65 (-1.02) |
> | SynWMD | 1-6 | SWF | SWD | 70.55 (-0.31) | 73.31 (-1.36) |
>
> These experimental results will be included in the camera-ready version. We appreciate your constructive comments.
>
> ---
>
> # Question about the evaluation dataset size
>
> > The evaluation dataset size is quite small.
>
> A2. We acknowledge your point about the small size of the evaluation dataset.
> **We intend to repeat the experiments** using the full PAWS dataset and other STS datasets such as STS12-16 and SICKR,  and include the results in the camera-ready version as long as time permits. Thank you again for your constructive feedback.
>
> To provide more context,  I explain the reason. The original dataset sizes for PAWSQQP, PAWSWiki, and STSB are as follows:
>
> |Dataset|Dev|Test|
> |---|---|---|
> |PAWSQQP|11988|677|
> |PAWSWiki|8000|8000|
> |STSB|1500|1379|
>
> Since the data sizes for the PAWSQQP and PAWSWiki test sets are significantly different in our study, we adjusted them for experimental efficiency. We adjusted the sizes of the PAWSQQP dev and PAWSWiki dev and test sets to the STSB dev size of 1500 because STSB is one of the most widely used datasets and we felt that the size of 1500 sentence pairs was not too small, although the PAWSQQP test set is relatively small.

---

### Meta-Review · Area_Chair_ZAK7 · 2023-09-22

**Recommendation:** 3

**Metareview:**

This paper extends the Word Mover's Distance (WMD), which measures sentence similarity, by incorporating structural information based on the self-attention matrix of encoder language models (e.g., BERT). Thus, the main contribution is a solid extension of an existing technique. Generally, this paper is well-written and easy to follow, although a Reviewer highlighted that some figures are hard to interpret. Despite the experiments being fairly extensive and based on significant sample sizes (as shown in the rebuttal), the results are mixed. In fact, the Authors report gains in paragraph identification but no change in performance in semantic textual similarity. Thus, the new method seems beneficial only in a subset of sentence similarity tasks. Based on the rebuttal, the Authors convincingly demonstrated that these findings hold true independent of the choice of backbone language model (although it remains uncertain how the method behaves at larger model scales). Potential limitations that remain to be fully addressed are 1) the fact that different encoders may rely on different vocabularies (as a result of different tokenizers), as the proposed solution (i.e., mapping) remains error-prone; 2) a thorough comparison with baselines for sentence similarity based on methods different from WMD, such as BERTScore, which already take into account structural information. The rebuttal has added some of these results to the initial submission; however, they remain incomplete and therefore no definitive conclusions can be drawn from them. Ultimately, I strongly recommend accepting this paper for Findings.

---

### Decision · Program_Chairs · 2023-10-07

**Decision:**

Accept-Findings

**Comment:**

This paper extends the Word Mover's Distance (WMD), which measures sentence similarity, by incorporating structural information based on the self-attention matrix of encoder language models (e.g., BERT). Thus, the main contribution is a solid extension of an existing technique. Generally, this paper is well-written and easy to follow, although a Reviewer highlighted that some figures are hard to interpret. Despite the experiments being fairly extensive and based on significant sample sizes (as shown in the rebuttal), the results are mixed. In fact, the Authors report gains in paragraph identification but no change in performance in semantic textual similarity. Thus, the new method seems beneficial only in a subset of sentence similarity tasks. Based on the rebuttal, the Authors convincingly demonstrated that these findings hold true independent of the choice of backbone language model (although it remains uncertain how the method behaves at larger model scales). Potential limitations that remain to be fully addressed are 1) the fact that different encoders may rely on different vocabularies (as a result of different tokenizers), as the proposed solution (i.e., mapping) remains error-prone; 2) a thorough comparison with baselines for sentence similarity based on methods different from WMD, such as BERTScore, which already take into account structural information. The rebuttal has added some of these results to the initial submission; however, they remain incomplete and therefore no definitive conclusions can be drawn from them. Ultimately, I strongly recommend accepting this paper for Findings.